# Community detection in sparse time-evolving graphs with a dynamical Bethe-Hessian

**Lorenzo Dall'Amico**
GIPSA-lab, UGA, CNRS, Grenoble INP
`lorenzo.dall-amico@gipsa-lab.fr`

**Romain Couillet**
GIPSA-lab, UGA, CNRS, Grenoble INP
L2S, CentraleSupélec, University of Paris Saclay

**Nicolas Tremblay**
GIPSA-lab, UGA, CNRS, Grenoble INP

## Abstract

This article considers the problem of community detection in sparse dynamical graphs in which the community structure evolves over time. A fast spectral algorithm based on an extension of the Bethe-Hessian matrix is proposed, which benefits from the positive correlation in the class labels and in their temporal evolution and is designed to be applicable to any dynamical graph with a community structure. Under the dynamical degree-corrected stochastic block model, in the case of two classes of equal size, we demonstrate and support with extensive simulations that our proposed algorithm is capable of making non-trivial community reconstruction as soon as theoretically possible, thereby reaching the optimal *detectability threshold* and provably outperforming competing spectral methods.

## 1 Introduction

Complex networks are a powerful tool to describe pairwise interactions among the members of a multi-agent system [1]. One of the most elementary tasks to be performed on networks is community detection [2, 3], *i.e.*, the identification of a non overlapping partition of the members (or nodes) of the network, representing its "mesoscale" structure. Although most of the attention is still focused on *static* community detection [3], many real networks are intimately dynamic: this is the case of networks representing physical proximity of mobile agents, collaboration interactions in the long run, biological and chemical evolution of group members, etc. (see [4] for a review).

There are many ways to define the concept of communities, particularly in dynamical networks (see *e.g.*, [3, 5]). In this article, we focus on the *dynamical degree corrected stochastic block model* (D-DCSBM), formally defined in Section 2, which is a variation of the static DCSBM [6, 7]. Specifically, letting $\mathcal{G}_t$ be the $k$-community graph at time instant $t$, $\mathcal{G}_t$ is generated independently of $\mathcal{G}_{t'}$ for all $t' \neq t$, but only a fraction $1 - \eta$ (for $\eta \in [0, 1]$) of the nodes changes class association between time $t$ and time $t + 1$. The degree correction lets the nodes have an arbitrary degree distribution, thereby possibly accounting for the broad distributions typical of real networks [8]. Of fundamental importance, in the static regime, two-class DCSBM graphs exhibit a *detectability threshold* below which no algorithm can asymptotically find a node partition non trivially aligned to the genuine classes [9, 10, 11, 12, 13]. Under a D-DCSBM model, one can similarly define a *dynamical detectability threshold* which considers the inference problem on the graph sequence $\{\mathcal{G}_t\}_{t=1,...,T}$ [14]. For $k > 2$ classes, the identification of a detectability threshold remains an open problem.

Spectral clustering is arguably one of the most successful ways to perform community detection [15]. Instances of spectral methods are indeed known to attain the detectability threshold in various

contexts (in dense [16, 17] or sparse [13, 18, 19, 20] stochastic block models) and are experimentally observed to perform competitively with the Bayes optimal solution [18, 19]. Recently, spectral clustering algorithms have also been explored in the dynamic regime [21, 22, 23, 24, 25, 26].

Two of the major pitfalls of dynamical spectral methods are *sparsity*, when the node degrees do not scale with the size $n$ of the graph, and *small label persistence*, when the fraction of nodes $1 - \eta$ that change label at any time instant $t$ is of order $O_n(1)$. Small persistence realistically assumes that, successive observations of the graph being independent across time, their community configuration must also evolve non-trivially. Under a sparse regime, but for $1 - \eta = o_n(1)$, [26] suggests to average the adjacency matrices over multiple time instances to obtain efficient community reconstruction. To the best of our knowledge, the work of [14] provides the only existing spectral algorithm properly treating both sparsity and small label persistence. In the spirit of [18], the proposed method arises from a linearization of the (asymptotically optimal) *belief propagation* algorithm (BP), which is capable of obtaining non-trivial partitions (*i.e.*, better than random guess) as soon as theoretically possible. However, their resulting dynamical non-backtracking matrix depends on an *a priori* unknown parameter[1], so the algorithm is practically inapplicable.

As an answer to these limitations, this article proposes a new spectral algorithm adapted to the sparse regime, which is able to detect communities even under little (or no) persistence in the community labels and which benefits from persistence to improve classification performance over a static algorithm run independently at each time-step. Specifically,

1. We introduce a dynamical Bethe-Hessian matrix which, for $k = 2$, retrieves non-trivial communities as soon as theoretically possible. As a by-product, we offer new results on the spectrum of the dynamical non-backtracking of [14].

2. We provide an algorithm applicable to any graph with $k \geq 2$ communities of arbitrary sizes.[2] On top of Python codes to reproduce most of the figures of this paper (available in the supplementary material), we provide an efficient Julia implementation, part of the CoDeBetHe package (community detection with the Bethe-Hessian), available at github.com/lorenzodallamico.

**Notations.** Function $\mathbb{1}_x$ is the indicator equal to 1 if condition $x$ is verified and 0 otherwise. Column vectors are indicated in bold ($\boldsymbol{v}$), matrices ($M$) and vector elements ($v_i$) in standard font. Vector $\mathbf{1}_n \in \mathbb{R}^n$ is the all-ones vector. The index $t$ always refers to time. The set $\partial i = \{j : (i, j) \in \mathcal{E}\}$ are the neighbors of $i$ in graph $\mathcal{G} = (\mathcal{V}, \mathcal{E})$ with edge set $\mathcal{E}$. The spectral radius of matrix $M$ is $\rho(M)$.

## 2  Model and setting

Let $\{\mathcal{G}_t\}_{t=1,\ldots,T}$ be a sequence of unweighted and undirected graphs, each with $n$ nodes. At time step $t$, $\mathcal{E}_t$ and $\mathcal{V}_t$ denote the set of edges and nodes, respectively, which form $\mathcal{G}_t$, with $\mathcal{V}_t \cap \mathcal{V}_{t'} = \emptyset$, for $t' \neq t$: each node has $T$ copies, each copy being a different object. We denote with $i_t$, for $1 \leq i \leq n$ and $1 \leq t \leq T$, a node in $\mathcal{V}_t$. We call $A^{(t)} \in \{0, 1\}^{n \times n}$ the symmetric adjacency matrix of $\mathcal{G}_t$, defined as $A_{ij}^{(t)} = \mathbb{1}_{(ij) \in \mathcal{E}_t}$, and $D^{(t)} = \mathrm{diag}(A^{(t)}\mathbf{1}_n) \in \mathbb{N}^{n \times n}$ its associated degree matrix. We now detail the generative model for $\{\mathcal{G}_t\}_{t=1,\ldots,T}$.

### 2.1  The dynamical degree corrected stochastic block model

For readability, until Section 4, where among other generalizations, we will consider graphs with an arbitrary number of classes $k$, we focus on a model with two classes of equal size. Let $\ell_{i_t} \in \{1, 2\}$ be the label of node $i_t$. The vector $\{\ell_{i_{t=1}}\}_{i=1,\ldots,n}$ is initialized by assigning random labels (1 or 2) with equal probability. The labels are then updated for $2 \leq t \leq T$ according to the Markov process

$$\ell_{i_t} = \begin{cases} \ell_{i_{t-1}} & \text{w.p. } \eta \\ a & \text{w.p. } \frac{1-\eta}{2}, \ \ a \in \{1, 2\}, \end{cases} \tag{1}$$

*i.e.*, the label of node $i_t$ is maintained with probability $\eta$ and otherwise reassigned at random with probability $1 - \eta$. Note that a proportion of the reassigned nodes from time $t$ will be affected the

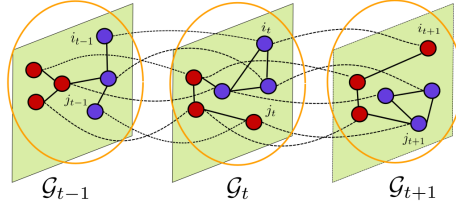

$$\mathcal{G}_{t-1} \qquad\qquad \mathcal{G}_{t} \qquad\qquad \mathcal{G}_{t+1}$$

Figure 1: Three successive instances of a dynamical network $\mathcal{G}$. Classes are emphasized by node colors and can evolve with time. Network edges, that change over time, are indicated in solid lines, while "temporal edges" in dashed lines connect each graph to its temporal neighbors. Nodes of a common time step are circled in orange.

same labels at time $t + 1$. The entries of the adjacency matrix $A^{(t)}$ of $\mathcal{G}_t$ are generated independently and independently across $t$, according to:

$$\mathbb{P}(A_{ij}^{(t)} = 1) = \theta_i \theta_j \frac{C_{\ell_{i_t}, \ell_{j_t}}}{n}, \quad \forall\, i > j. \tag{2}$$

The vector $\boldsymbol{\theta} = (\theta_1, \ldots, \theta_n)$ enables to induce any arbitrary degree distribution and satisfies $\frac{1}{n}\sum_{i=1}^{n} \theta_i = 1$ and $\frac{1}{n}\sum_{i=1}^{n} \theta_i^2 \equiv \Phi = O_n(1)$. The matrix $C \in \mathbb{R}^{2 \times 2}$ contains the class affinities with $C_{a=b} \equiv c_{\text{in}}$ and $C_{a \neq b} \equiv c_{\text{out}}$, $c_{\text{in}}$ and $c_{\text{out}}$ being independent of $n$. The expected average graph degree is $c \equiv (c_{\text{in}} + c_{\text{out}})/2 = O_n(1)$ assumed to satisfy $c\Phi > 1$: according to (1)–(2), this is the necessary (and sufficient) condition such that, at each time step, $\mathcal{G}_t$ has a giant component[3] [27]. This condition imposes constraint on $c$, hence on how sparse the graphs $\{\mathcal{G}_t\}_{t=1,\ldots,T}$ can be.

We insist that the process (1)–(2) builds on a dual time-scale assumption: a short range governing the evolution of graph edges (reconfigured at each time step) and a long range governing the evolution of communities. The article mainly focuses on the long range evolution as independent realizations of $\mathcal{G}_t$ are assumed at successive times. Appendix D discusses the extension of this framework to $\mathcal{G}_t$ evolving slowly with time, thereby allowing for *edge persistence* across time.

Our objective is to solve the problem of community reconstruction on the *dynamical graph* $\mathcal{G}$ constructed, as illustrated in Figure 1, from the $T$ independent instances $\{\mathcal{G}_t\}_{t=1,\ldots,T}$.

**Definition 1** *Letting $\{\mathcal{G}_t\}_{t=1\ldots T}$ be a sequence of graphs independently generated from (1)–(2), $\mathcal{G} = \mathcal{G}(\mathcal{V}, \mathcal{E})$ is the graph with $\mathcal{V} = \cup_{t=1}^{T} \mathcal{V}_t$ and $\mathcal{E} = \left(\cup_{t=1}^{T} \mathcal{E}_t\right) \cup \left(\cup_{t=1}^{T-1} \cup_{i=1}^{n} (i_t, i_{t+1})\right)$. The adjacency and degree matrices of $\mathcal{G}$ are denoted with $A, D \in \mathbb{N}^{nT \times nT}$, respectively. In other words, the graphs $\mathcal{G}_t$ are joined adding extra edges between the nodes $i_t$ and their temporal neighbors $i_{t\pm1}$.*

## 2.2 Detectability threshold in the D-DCSBM

Let $\lambda = (c_{\text{in}} - c_{\text{out}})/(c_{\text{in}} + c_{\text{out}})$ be the co-variance between neighboring labels [13, 28]. Based on [29], the authors of [14] conjecture that, for the D-SBM (for which $\theta_i = 1$ for all $i$), as $n, T \to \infty$, non-trivial class reconstruction is feasible if and only if $\alpha \equiv \sqrt{c\lambda^2} > \alpha_c(\infty, \eta)$, where $\alpha_c(\infty, \eta)$ is the *detectability threshold* defined as the unique value of $\bar{\alpha} > 0$ for which the largest eigenvalue of

$$M_\infty(\bar{\alpha}, \eta) = \begin{pmatrix} \bar{\alpha}^2 & 2\eta^2 \\ \bar{\alpha}^2 & \eta^2 \end{pmatrix} \tag{3}$$

is equal to one. Inspired by [13] who adapted the detectability condition to the DCSBM model in the static case, we show (see Appendix A) that this result can be extended to the D-DCSBM by (i) redefining $\alpha$ as $\alpha \equiv \sqrt{c\Phi\lambda^2}$ and (ii) for finite $T$ (but $n \to \infty$) by redefining $\alpha_c(T, \eta)$ as the value

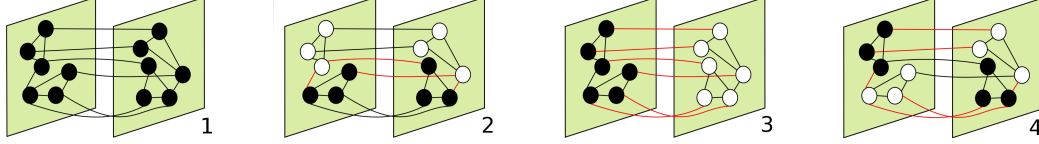

Figure 2: Sketch of the 4 stable modes for two communities and $T = 2$. In black we indicate the direction $s_i > 0$, in white $s_i < 0$. The red edges correspond to the frustrated edges connecting spins with opposite direction.

of $\bar{\alpha}$ for which the largest eigenvalue of

$$M_T(\bar{\alpha}, \eta) = \begin{pmatrix} M_d & M_+ & 0 & \dots & 0 \\ M_- & M_d & \ddots & \dots & 0 \\ 0 & M_- & \ddots & M_+ & 0 \\ \vdots & \vdots & \ddots & M_d & M_+ \\ 0 & 0 & \dots & M_- & M_d \end{pmatrix}, \text{ where } \begin{cases} M_d = \begin{pmatrix} 0 & 0 & 0 \\ \eta^2 & \bar{\alpha}^2 & \eta^2 \\ 0 & 0 & 0 \end{pmatrix}, \\ M_+ = \begin{pmatrix} 0 & 0 & 0 \\ 0 & 0 & 0 \\ 0 & \bar{\alpha}^2 & \eta^2 \end{pmatrix}, \\ M_- = \begin{pmatrix} \eta^2 & \bar{\alpha}^2 & 0 \\ 0 & 0 & 0 \\ 0 & 0 & 0 \end{pmatrix}, \end{cases} \quad (4)$$

is equal to one. The detailed derivation of $M_T(\bar{\alpha}, \eta)$ are reported in Appendix A, which provides an explicit expression to $\alpha_c(T, \eta)$, following the arguments of [14]. The definition of $M_T(\bar{\alpha}, \eta)$ is more elaborate than $M_\infty(\bar{\alpha}, \eta)$ due to the finite-time structure of $\mathcal{G}$: each node $i_t$ has two temporal connections with $i_{t+1}$ and $i_{t-1}$, except for the "time boundary" nodes of $\mathcal{G}_{t=1}$ and $\mathcal{G}_{t=T}$. As $T \to \infty$, these boundaries can be neglected and the leading eigenvalue of $M_T(\bar{\alpha}, \eta)$ reduces to that of $M_\infty(\bar{\alpha}, \eta)$. The expression of $\alpha_c(T, \eta)$ can be computed analytically for $T = 2, 3, 4$ and $T \to \infty$:

$$\alpha_c(T = 2, \eta) = \left(1 + \eta^2\right)^{-\frac{1}{2}}; \qquad \alpha_c(T = 3, \eta) = \sqrt{2}\left(2 + \eta^4 + \eta^2\sqrt{8 + \eta^4}\right)^{-\frac{1}{2}} \quad (5)$$

$$\alpha_c(T = 4, \eta) = \sqrt{2}\left(2 + \eta^2 + \eta^6 + \eta\sqrt{\eta^8 + 2\eta^4 + 8\eta^2 + 5}\right)^{-\frac{1}{2}}; \quad \alpha_c(\infty, \eta) = \left(\frac{1 + \eta^2}{1 - \eta^2}\right)^{-\frac{1}{2}}.$$

For other values of $T$, $\alpha_c(T, \eta)$ is best evaluated numerically. For all $T$: (i) if $\eta = 0$ (no correlation among the labels), one recovers $\alpha_c = 1$, the transition's position in the static DCSBM [13], as expected; (ii) if $\eta = 1$, $\alpha_c = 1/\sqrt{T}$, the static threshold obtained by averaging the adjacency matrix over its $T$ independent and identically distributed realizations. We also numerically confirm that for all $T$, $\alpha_c(T, \eta)$ is a decreasing function of $\eta$: higher label persistence allows to solve harder problems.

## 3 Main results

This section develops a new "dynamical" Bethe-Hessian matrix associated to the graph $\mathcal{G}$, for which we show there exists *at least* one eigenvector (recall that $k = 2$ classes so far) strongly aligned to the community labels if $\alpha > \alpha_c(T, \eta)$, thereby allowing for high performance community detection down to the detectability threshold. The eigenvectors containing information can be up to $T$, but only one of them is guaranteed to exist when $\alpha > \alpha_c(T, \eta)$ and it can alone reconstruct communities.

### 3.1 The dynamical Bethe-Hessian matrix

As in [19, 28], our approach exploits a statistical physics analogy between the modelling of spontaneous magnetization of spins with *ferromagnetic* interaction [30] and the modelling of communities of nodes in sparse graphs. We attach to each node a *spin* variable $s_{i_t} \in \{\pm 1\}$, for $1 \le i \le n$ and $1 \le t \le T$. The energy of a spin configuration $\boldsymbol{s} \in \{\pm 1\}^{nT}$ is given by the *Hamiltonian*

$$\mathcal{H}_{\xi,h}(\boldsymbol{s}) = -\sum_{t=1}^{T}\left(\sum_{(i_t, j_t) \in \mathcal{E}_t} \text{ath}(\xi)\, s_{i_t} s_{j_t} + \sum_{i_t \in \mathcal{V}_t} \text{ath}(h)\, s_{i_t} s_{i_{t+1}}\right) \quad (6)$$

with $s_{i_{T+1}} = 0$ by convention. Here, the coupling constants $\xi, h \in [0, 1)$ modulate the interaction among nodes at time $t$ and between the same node at time instants $t$ and $t + 1$, respectively, and

appear inside inverse hyperbolic tangents for notational ease. Intuitively, the spin vector $s$ can be mapped to the class affiliation vector $\sigma = 2\ell - 3$. The first term in the main parenthesis of (6) favors configurations in which neighboring nodes have the same label, while the second term favors configurations in which the label is kept across successive time instants. This last term enforces persistence in the community evolution.

The configurations $s$ representing the local minima of $H_{\xi,h}(s)$ are determined by the mesoscale structure of $\mathcal{G}$ and are sketched for $T = 2$ in Figure 2. The lowest energy state corresponds to $s = 1_{nT}$: this is the non-informative *ferromagnetic configuration*. Similarly, mode 3 of Figure 2 groups together nodes in different communities and is equally useless for reconstruction. On the opposite, modes 2 and 4 of Figure 2 divide the nodes according to the class structure of $\mathcal{G}$ and can be used for community reconstruction. In general, for $k$ classes and $T > 2$ time frames, $kT$ local minima arise, mixing together time and class clusters. Note importantly that mode 1 always has a lower energy than mode 3 and mode 2 a lower energy than mode 4. However, the ordering of energies of modes 2 and 3 is in general not *a priori* known. We will further comment on this remark which has important consequences for the subsequent analysis as well as for the design of our proposed community detection algorithm.

We show in Appendix B that these lowest energy modes can be approximated by the eigenvectors associated with the smallest eigenvalues of the *Bethe-Hessian* matrix $H_{\xi,h} \in \mathbb{R}^{nT \times nT}$, defined by

$$(H_{\xi,h})_{i_t, j_{t'}} = \begin{cases} \left( \frac{\xi^2 D^{(t)} - \xi A^{(t)}}{1 - \xi^2} + \frac{1 + h^2(\phi_t - 1)}{1 - h^2} I_n \right)_{ij} & \text{if } t = t' \\ \left( -\frac{h}{1 - h^2} I_n \right)_{ij} & \text{if } t = t' \pm 1, \end{cases} \tag{7}$$

in which $\phi_t = 1$ if $t = 1$ or $t = T$ and $\phi_t = 2$ otherwise. The aforementioned lack of a precise knowledge of the relative position of the informative modes in the energy spectrum of the Hamiltonian hampers the identification of the position of the corresponding informative eigenvectors of $H_{\xi,h}$. This is of major importance when designing a spectral clustering algorithm based on $H_{\xi,h}$.

## 3.2 Community detectability with the dynamic Bethe-Hessian

We thus now turn to our main result (Proposition 1), whose theoretical support is given in Appendix C, centered on the question of appropriately choosing a pair $(\xi, h)$ which ensures non-trivial community detection with $H_{\xi,h}$ as soon as $\alpha > \alpha_c(T, \eta)$ and which, in addition, necessarily exploits the informative eigenvectors of $H_{\xi,h}$ without knowing their precise location in the spectrum.

Let us first introduce an important intermediary object: the weighted non-backtracking matrix $B_{\xi,h}$, defined on the set of *directed* edges $\mathcal{E}^d$ of $\mathcal{G}$. Letting $\omega_{ij} = \xi$ if there exists a time instant $t$ such that nodes $i, j$ belong to $\mathcal{V}_t$, and $\omega_{ij} = h$ for time edges, the entries $(ij), (kl) \in \mathcal{E}^d$ of $B_{\xi,h}$ are defined as

$$(B_{\xi,h})_{(ij)(kl)} = \mathbb{1}_{jk}(1 - \mathbb{1}_{il}) \, \omega_{kl}. \tag{8}$$

The spectra, and notably the isolated eigenvalues and their associated eigenvectors, of the matrices $B_{\xi,h}$ and $H_{\xi,h}$ have important common properties [31, 32]. As $n \to \infty$, both the spectra of the Bethe-Hessian and non-backtracking matrices are the union of isolated eigenvalues (the eigenvectors of which carry the information on the mesoscale structure of $\mathcal{G}$) and of a bulk of uninformative eigenvalues [20, 12]. This relation allows us to establish the following key result.

**Proposition 1** *Let $\lambda_d = \frac{\alpha_c(T,\eta)}{\sqrt{c\Phi}}$. Then, as $n \to \infty$, (i) the complex eigenvalues forming the bulk spectrum of $B_{\lambda_d,\eta}$ are asymptotically bounded within the unit disk (ii) the smallest eigenvalues of the (real) bulk spectrum of $H_{\lambda_d,\eta}$ tend to $0^+$ and (iii) the number of isolated negative eigenvalues of $H_{\lambda_d,\eta}$ is equal to the number of real isolated eigenvalues of $B_{\lambda_d,\eta}$ greater than 1.*

*In particular, if $\alpha > \alpha_c(T, \eta)$, at least one of the isolated real eigenvalues of $B_{\lambda_d,\eta}$ larger than 1 and one of the negative isolated eigenvalues of $H_{\lambda_d,\eta}$ are informative in the sense that their associated eigenvectors are correlated to the vector of community labels.*

Proposition 1 indicates that, if $\alpha > \alpha_c(T, \eta)$, certainly there is one informative eigenvector (more precisely, mode 2 of Figure 2) which is associated with *one of the few isolated negative eigenvalues*

of $H_{\lambda_d,\eta}$. Other informative eigenvectors (*e.g.* mode 4 of Figure 2) may be associated to negative eigenvalues of $H_{\lambda_d,\eta}$, but their existence is not guaranteed. By performing spectral clustering on these few negative eigenvalues and appropriately handling the size-$nT$ eigenvectors, one can then be assured to extract the desired community information. We empirically confirm that using *all* the eigenvectors associated with the isolated negative eigenvalues (instead of only the desired informative eigenvector with unknown location) to form a *low dimensional vector embedding* of the nodes is redundant but it does not severely compromise the performance of the final k-means step of the standard spectral clustering method [33]. The choice $\xi = \lambda_d$ and $h = \eta$ therefore almost immediately induces an explicit algorithm applicable to arbitrary networks and which, as later discussed in Section 4, straightforwardly extends to graphs with $k > 2$ communities.

To best understand the structure of $H_{\lambda_d,\eta}$, a further comment should be made on the expected number of its negative eigenvalues. It may in particular be shown that, in the limit $\eta \to 0$, the off-diagonal blocks of $H_{\lambda_d,\eta}$ vanish and exactly $2T$ negative eigenvalues get isolated, the $T$ smallest negative being almost equal and uninformative and the latter $T$ almost equal but informative. In the limit $\eta \to 1$ instead, the configurations alike modes 3 and 4 of Figure 2 are energetically penalized (recall (6)) and do not produce any isolated eigenvalue, thus $H_{\lambda_d,\eta}$ only has two negative eigenvalues.

Appendix C shows that a better choice for $\xi$ is in fact $\lambda$, instead of $\lambda_d$. Experimental verification confirms that, as in the static regime [28], this is due to the fact that, unlike $H_{\lambda,\eta}$, the entries of the informative eigenvectors of $H_{\lambda_d,\eta}$ are tainted by the graph degrees, thereby *distorting to some extent* the class information.[4] On the opposite, the eigenvector of $H_{\lambda,\eta}$ associated to the eigenvalue closest to zero (which in this case is isolated while the bulk is away from zero) is informative but *not tainted* by the graph degree heterogeneity. Although both choices of $\xi$ provably enable non-trivial community recovery down to the threshold, $\xi = \lambda$ is expected to outperform $\xi = \lambda_d$, especially as $\alpha$ increases away from the threshold. Consequently, if one has access to prior knowledge on $\lambda$, then the eigenvectors of $H_{\lambda,\eta}$ should be used for best performance. However, in practice, providing a good estimate of $\lambda$ in reasonable time remains a challenge, especially for $k \geq 2$. This is why we prefer the choice $\xi = \lambda_d$, as $\lambda_d$ is an explicit function of $\alpha_c(T,\eta)$, $c$ and $\Phi$ all of which can be easily estimated.

## 4  Algorithm and performance comparison

These discussions place us in a position to provide an algorithmic answer to the dynamic community detection problem under study. The algorithm, Algorithm 1, is shown here to be applicable, up to a few tailored adjustments, to arbitrary real dynamical graphs.

### 4.1  Algorithm implementation on arbitrary networks

We have previously summarized the main ideas behind a dynamical version of spectral clustering based on $H_{\lambda_d,\eta}$. These form the core of Algorithm 1. Yet, in order to devise a practical algorithm, applicable to a broad range of dynamical graphs, some aspects that go beyond the D-DCSBM assumption should be taken into account.

So far, the article dealt with $k = 2$ equal-size communities for which the D-DCSBM threshold is well defined. Real networks may of course have multiple asymmetrical-sized classes. As in the static case [20], we argue that, under this general D-DCSBM setting and the classical assumption that the expected degree of each node is class-independent, the left edge of the bulk spectrum of $H_{\lambda_d,\eta}$ is still asymptotically close to zero and that some of the eigenvectors associated with the isolated negative eigenvalues carry information for community reconstruction.[5] The value $k$ is, in practice, also likely unknown. This also does not affect the idea of the algorithm which exploits all eigenvectors associated to the negative eigenvalues of $H_{\lambda_d,\eta}$, without the need of knowing $k$. The very choice of $k$ is only required by k-means in the last step of spectral clustering and may be performed using off-the-shelf k-means compliant tools, *e.g.*, the *silhouettes method* [34].

**Algorithm 1** Community detection in sparse, heterogeneous and dynamical graphs

---
1: **Input** : adjacency matrices $\{A^{(t)}\}_{t=1,\dots,T}$ of the undirected graphs $\{\mathcal{G}_t\}_{t=1,\dots,T}$; label persistence, $\eta$; number of clusters $k$.
2: **for** $t = 1 : T - 1$ **do**
3:     Remove from $A^{(t+1)}$ the edges appearing in both $A^{(t)}$ and $A^{(t+1)}$ (Appendix D)
4: Compute: $d_i^{(t)} \leftarrow \sum_{j=1}^{n} A_{ij}^{(t)}$; $c \leftarrow \frac{1}{nT} \sum_{t=1}^{T} \sum_{i=1}^{n} d_i^{(t)}$; $\Phi \leftarrow \frac{1}{nTc^2} \sum_{t=1}^{T} \sum_{i=1}^{n} \left( d_i^{(t)} \right)^2$;
    $\alpha_c(T, \eta)$ from Equation (4); $\lambda_d \leftarrow \frac{\alpha_c(T,\eta)}{\sqrt{c\Phi}}$ .
5: Stack the $m$ eigenvectors of $H_{\lambda_d,\eta}$ with negative eigenvalues in the columns of $X \in \mathbb{R}^{nT \times m}$
6: Normalize the rows of $X_{i,:} \leftarrow X_{i,:} / \|X_{i,:}\|$
7: **for** $t = 1 : T$ **do**
8:     Estimate the community labels $\{\hat{\ell}_{i_t}\}_{i=1,\dots n}$ using $k$-class *k-means* on the rows $\{X_{i_t}\}_{i=1,\dots,n}$.
9: **return** Estimated label vector $\hat{\boldsymbol{\ell}} \in \{1, \dots, k\}^{nT}$.

---

Another aspect of practical concern is that successive realizations of $A^{(t)}$ may not be independent across time. Appendix D, covers this issue by introducing *edge persistence* in the model. As suggested in [35], by simply removing from $A^{(t+1)}$ all edges also present in $A^{(t)}$, one then retrieves a sequence of adjacency matrices which, for sparsity reasons, (asymptotically) mimic graphs without edge dependence. These updated adjacency matrices are a suited input replacement to the algorithm.

A last important remark is that $\eta$ is an input of Algorithm 1. If unknown, as it would in general be, one may choose an arbitrary $h \in [0, 1)$ and $\xi = \alpha_c(T, h)$, to then perform spectral clustering on $H_{\xi,h}$: the leftmost edge of the bulk spectrum of $H_{\xi,h}$ is asymptotically close to zero for all $h$ and consequently Algorithm 1 can be used in the same form. However, for a mismatched $h$, the detectability threshold now occurs beyond the optimal $\alpha_c(T, \eta)$. Close to the transition, this mismatch would give rise to fewer informative isolated negative eigenvalues than expected, resulting in a poor quality label assignment. As a workaround, one may browse through a discrete set of values for $h$ and extract the $h$ maximizing some quality measure, such as the resulting clustering *modularity*. [36].

**Computation complexity.** The bottleneck of Algorithm 1 is to compute the embedding $X$. The number of negative eigenvalues $m$ is not *a priori* known and only suspected to be in the interval $\{k, \dots, kT\}$. Our strategy is to compute the first $k + 1$ eigenvectors, ensure that the associated eigenvalues are all negative, then compute the $(k + 2)$-th eigenvector, etc., until the largest uncovered eigenvalue crosses zero. This strategy, via standard sparse numerical algebra tools based on Krylov subspaces [37], costs $\mathcal{O}(nT \sum_{l=k}^{m} l^2)$. In the best-case (resp., worst-case) scenario, $m = k$ (resp., $m = kT$): the complexity of Algorithm 1 thus scales as $\mathcal{O}(nTk^2)$ (resp., $\mathcal{O}(nT^4k^3)$).

**An accelerated approximate implementation.** As $T$ or $k$ increase, the above complexity may become prohibitive. A recent workaround strategy [38, 39, 40], based on polynomial approximation and random projections, is here particularly adapted, and decreases the overall complexity of the algorithm to $\mathcal{O}(nTk \log(nT))$, for a limited loss in precision. The resulting fast implementation is described in Algorithm 2 and detailed in Appendix F. To give an order of magnitude, a simulation[6] of Algorithm 1 for $n = 10^5$, $T = 5$ (resp., $n = 5\,000$, $T = 100$), $k = 2$, $c = 6$, $\eta = 0.5$, $\Phi = 1.6$, $\alpha = 2\alpha_c(T, \eta)$ takes on average approximately 1 minute (resp., 40 minutes), whereas Algorithm 2 converges in less than 4 minutes in both cases. The reader is referred to Appendix F for more details.

## 4.2 Performance comparison on synthetic datasets

Figure 3 shows the performance of different clustering algorithms in terms of overlap

$$\mathrm{ov}(\boldsymbol{\ell}, \hat{\boldsymbol{\ell}}) = \max_{\bar{\boldsymbol{\ell}} \in \mathcal{P}(\hat{\boldsymbol{\ell}})} \frac{1}{1 - \frac{1}{k}} \left( \frac{1}{n} \sum_{i=1}^{n} \mathbb{1}_{\ell_i, \bar{\ell}_i} - \frac{1}{k} \right), \tag{9}$$

where $\boldsymbol{\ell}, \hat{\boldsymbol{\ell}} \in \{1, \dots, k\}^n$ are the ground truth and estimated label vectors, respectively, while $\mathcal{P}(\boldsymbol{\ell})$ is the set of permutations of $\boldsymbol{\ell}$. The overlap ranges from zero for a random label assignment to one for perfect label assignment. Figure 3-left compares the overlap performance as a function of $\alpha$ and $\eta$ for

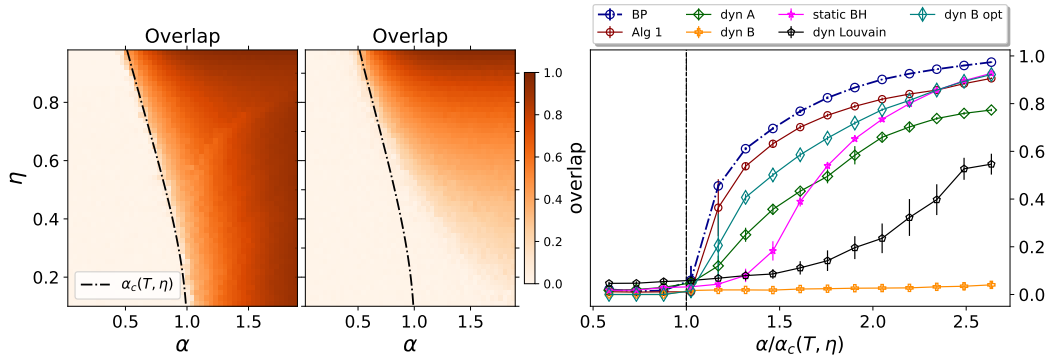

Figure 3: **Left**: overlap comparison at $t = T$ for Algorithm 1 vs. [26], in color gradient, for various detectability hardness levels $\alpha$ ($x$-axis) and label persistence $\eta$ ($y$-axis); $n = 10\,000$, $T = 5$, $c = 10$, $\Phi = 1$; averaged over 4 samples. **Right**: mean overlap across all values of $t$, as a function of $\alpha$, for Algorithm 1 (Alg 1), BP [14], the dynamic adjacency matrix of [26] (dyn A), the dynamical non-backtracking of [14] (dyn B and dyn B opt), the static Bethe-Hessian of [28] (static BH) and the dynamical Louvain algorithm of [41] (dyn Louvain); $n = 5\,000$, $T = 4$, $c = 6$, $\eta = 0.7$, $\Phi = 1$; averaged over 20 samples (3 for BP). For all plots, $k = 2$.

Algorithm 1 versus the adjacency averaging method of [26] (which we recall assumes $\eta = 1 - o_n(1)$). The overlap is only considered at $t = T$ so to compare Algorithm 1 on even grounds with [26] which only outputs one partition (rather than one for every $t$). The theoretical detectability threshold line $\alpha = \alpha_c(T, \eta)$ visually confirms the ability of Algorithm 1 to assign non trivial class labels as soon as theoretically possible, as opposed to the method of [26] which severely fails at small values of $\eta$.

Figure 3-right then compares the average overlap performance of Algorithm 1 against competing methods, for varying detection complexities $\alpha/\alpha_c(T, \eta)$. Algorithm 1 is outperformed only by the BP algorithm[7], but has an approximate 500-fold reduced computational cost. The computational heaviness of BP becomes practically prohibitive for larger values of $n$. For completeness, Appendix E provides further numerical performance comparison tests for different values of $\eta$, $\Phi$, for $k > 2$, for larger values of $n$ and $T$, and for graphs with clusters of different average sizes. Interestingly, for large values of $\alpha$, Algorithm 1 is slightly outperformed by the static Bethe-Hessian of [28], independently run at each time-step. As discussed at the end of Section 3, the choice $\xi = \lambda_d$ is sub-optimal compared to the optimal (but out-of-reach in practice) choice $\xi = \lambda$, the difference becoming more visible as $\alpha$ increases away from $\alpha_c$. Supposing one has access to an oracle for $\lambda$, running Algorithm 1 on $H_{\lambda,\eta}$ outputs a performance in terms of overlap (not shown) that is first super-imposed with the "Alg 1" plot for small values of $\alpha$ and gradually converges to the performance of "BP" as $\alpha$ increases; thus outperforming "static BH" everywhere. From a dynamical viewpoint, also, the large $\alpha$ regime is of least importance as a static algorithm can, alone, output a perfect reconstruction. Further numerical experiments are shown in Appendix E.

For the non-backtracking method of [14] ("dyn B"), the authors suggest to use (as we did here) the eigenvector associated to the second largest eigenvalue of $B_{\lambda,\eta}$, which, as $H_{\lambda_d,\eta}$, may also have informative and uninformative eigenvalues in reversed order. The curve "dyn B opt" shows the performance obtained using all the isolated eigenvectors of $B_{\lambda,\eta}$ and it confirms – in agreement with Appendices C and E and the claims of [14] – that $B_{\lambda,\eta}$ can indeed make non-trivial community reconstruction for all $\alpha > \alpha_c(T, \eta)$. Note that, as in the static case [18, 19], $B_{\lambda,\eta}$ is outperformed by $H_{\lambda_d,\eta}$ which, additionally, is symmetric and smaller in size, is well defined regardless of $\lambda$ and is, therefore, a more suitable candidate for community detection.

### 4.3 Test on Sociopatterns *Primary school*

This section shows the results of our experiments on the *Primary school* network [42, 43] of the SocioPatterns project. The dataset contains a temporal series of contacts between children and teachers

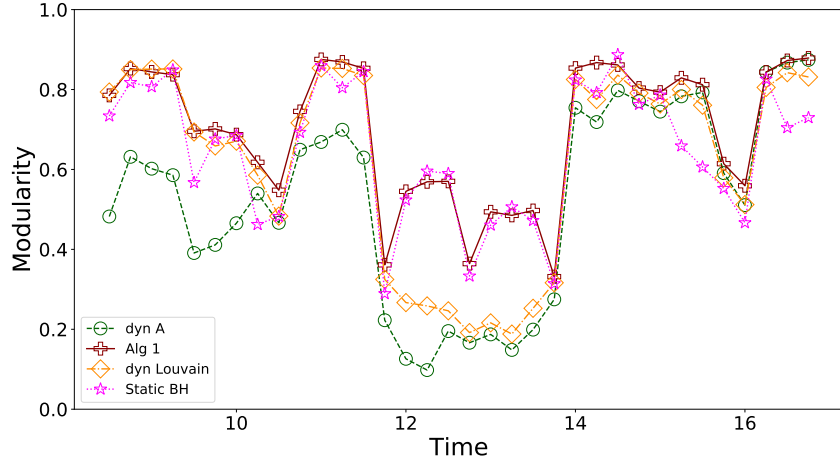

Figure 4: Modularity as a function of time for Algorithm 1 (Alg 1) for $\eta = 0.55$, the dynamic adjacency matrix of [26] (dyn A), the dynamic Louvain algorithm [41] (dyn Louvain) and the static Bethe-Hessian of [28] (static BH). The graph $\{\mathcal{G}_t\}_{t=1,...,T}$ is obtained from the *Primary school* network [42, 43] dataset, as in Section 4.3. For Algorithm 1, [28] and [26], $k = 10$ is imposed.

of ten classes of a primary school. For each time $1 \leq t \leq T$, $\mathcal{G}_t$ is obtained considering all interactions from time $t$ to time $t + 15$ min, starting from $t_1 = 8{:}30$ am until $t_T = 5$ pm for $T = 33$. Figure 4 compares the modularity as a function of time for different clustering techniques. We empirically observe that, for this dataset, multiple values of $\eta$ give similar results: this is not surprising because the clusters are here well delineated and we are in the (less interesting) easy detection regime. The value $\eta = 0.55$ is considered as an input of Algorithm 1, because it approximately matches the value of $\eta$ estimated from the inferred label vector $\hat{\ell}$ (see Equation (1)).

Figure 4 shows that Algorithm 1 is better than [26, 41] at all times, with a drastic gain during the lunch break, in which the community structure is harder to delineate. As compared to the static Bethe-Hessian, Algorithm 1 is slightly outperformed only on some times during the lunch break, while for other times it benefits from the positive correlation of the labels. Defining a unique, time independent $\eta$ certainly hampers the performance on this specific dataset in which a very large $\eta$ is expected during the lesson times, while a small $\eta$ may be more appropriate during the lunch break.

## 5   Concluding remarks

By means of arguments at the crossroads between statistical physics and graph theory, this article tailored Algorithm 1, a new spectral algorithm for community detection on sparse dynamical graphs. Algorithm 1 is capable of reconstructing communities as soon as theoretically possible, thereby largely outperforming state-of-the-art competing spectral approaches (especially when classes have a short-term persistence) while only marginally under-performing the (theoretically claimed optimal but computationally intensive) belief propagation algorithm.

A delicate feature of Algorithm 1 concerns the estimation of the class-persistence parameter $\eta$, if not available. We hinted in Section 4 at a greedy line-search solution which is however computationally inefficient and lacks of a sound theoretical support. This needs be addressed for Algorithm 1 to be more self-contained and applicable to the broadest range of practical networks.

Beyond this technical detail, the present analysis only scratches the surface of dynamical community detection: the problem in itself is vast and many degrees of freedom have not been here accounted for. The label persistence $\eta$ and community strength matrix $C$ (and thus the parameter $\lambda$ in a symmetric two-class setting) are likely to evolve with time as well. We empirically observed that Algorithm 1 naturally extends to this setting, each temporal block of the matrix $H_{\cdot,\cdot}$ now using its corresponding $\lambda_d^{(t)}$ and $\eta_t$. Yet, while Algorithm 1 seems resilient to a more advanced dynamical framework, the very concept of *detectability thresholds* becomes more elusive in a symmetrical two-class setting: a proper metric to measure the distance to optimality would thus need to be first delineated.

## Broader impact

Community detection algorithms have a broad interest as they can be applied to a very vast class of problems and settings. An interesting example, of utmost importance in the present days, was given by [44] were the authors showed the importance of keeping track of the time-evolving community structure of social networks to properly model an epidemic spreading. Not unlike any other clustering algorithm, however, when applied to a real social network, our algorithm can potentially evidence differences in terms of *e.g.* race, sex, religion. As discussed in [45], if such an output is used in some decision process, the result can indeed produce discriminatory choices.

Although we are aware of the potential weaknesses, the mainly theoretical nature of our study, as well as the nowadays vast literature in the field of community detection, allows us to not foresee any major negative consequence from our study. On the contrary, keeping into account of the realistic time-evolving nature of networks can allow to improve and better understand our studies in the field.

## Acknowledgements

RC's work is supported by the MIAI LargeDATA Chair at University Grenoble-Alpes and the GIPSA-HUAWEI Labs project Lardist. NT's work is partly supported by the French National Research Agency in the framework of the "Investissements d'avenir" program (ANR-15-IDEX-02) and the LabEx PERSYVAL (ANR-11-LABX-0025-01).

## Footnotes

[1]In order to design their dynamical non-backtracking matrix, the average number of connections among nodes in the same and across communities must be known.

[2]The algorithm *a priori* requires that $\eta$ be known; otherwise, $\eta$ can be estimated through cross-validation.

[3]The existence of a giant component at each time $t$ ensures a well-defined community detection problem when $n \to \infty$. In practice, $\mathcal{G}_t$ will typically be the union of a giant connected sub-graph, in which all communities are represented, and a few isolated nodes. These isolated nodes can be understood as nodes of a network absent at time $t$. In this sense, the D-DCSBM is suitable to model dynamic networks with varying size across time.

[4]In the present symmetric $k = 2$ setting, one expects the entries of the informative eigenvector to be noisy versions of $\pm 1$ values in which the degree dependence intervenes only in the variance, but not in the mean (see [28, 27] for a thorough study in the static case). For $\xi = \lambda_d$ though, the mean itself depends on the node degree and impedes the performance of k-means.

[5]In passing, while $\alpha_c(T,\eta)$ is well defined for all $k \geq 2$, when $k > 2$, its value no longer corresponds to the position of a detectability threshold, the very notion of which remains an open riddle for $k > 2$.

[6]The laptop's RAM is 7.7 Gb with Intel Core i7-6600U CPU @ 2.6GHz x 4.

[7]The codes used to obtain the BP performance displayed in Figure 3 are courtesy of Amir Ghasemian.

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
