[Supplementary Material · app.pdf]

# Supplementary material

The supplementary material provides complementary technical arguments to the main results of the article (Sections A–C), along with a discussion on the extension of the present setting to dynamic graphs with *link persistence* across time (Section D). Further numerical tests on the performance of Algorithm 1 are presented in Section E, while Section F presents the detailed description of Algorithm 2 to handle fast approximate spectral clustering.

## A Detectability threshold for finite $T$

This section discusses the conjecture of [14] in which the authors introduce a threshold $\alpha_c(T, \eta)$ (however not explicitly defined[8]), below which ($\alpha < \alpha_c(T, \eta)$) community detection is not feasible. We go here beyond [14] by providing an explicit value for $\alpha_c(T, \eta)$ for all finite $T$.

As a consequence of the sparsity of each $\mathcal{G}_t$, the graph $\mathcal{G}$, obtained by connecting together the same node at successive times as per Definition 1 (recall Figure 1) is locally tree-like, *i.e.* the local structure of $\mathcal{G}$ around a node $v \in \mathcal{V}$ is the same as that of a Galton-Watson tree $\mathcal{T}(v)$ [46], rooted at $v$, designed according to the following procedure: let $\ell_v \in \{1, 2\}$ be the label of $v$; next generate its progeny by creating $d_s$ spatial children (*i.e.*, nodes which live at the same time as $v$), where $d_s$ is a Bernoulli random variable with mean $c\Phi$, and two temporal children (*i.e.*, nodes which are the projection of $v$ at neighbouring times); for each spatial child $w$, assign the label $\ell_w = \ell_v$ with probability $c_{\text{in}}/(c_{\text{in}} + c_{\text{out}})$ and $\ell_w = 3 - \ell_v$ otherwise; the temporal children keep the same label as $v$ with probability $(1 + \eta)/2$ and change it with probability $(1 - \eta)/2$; each node thus created further generates its own set of offspring, with the only difference that the temporal children only bear one extra temporal child, while spatial children bear two.

In the limit $n, T \to \infty$, for any arbitrary $v \in \mathcal{V}$, the local structure of $\mathcal{G}$ around $v$ is the same as $\mathcal{T}(v)$, the Galton-Watson tree rooted at $v$. This means that, within a neighborhood reachable in a finite number of steps from $v$ in $\mathcal{G}$ or $\mathcal{T}(v)$, the probability distribution of the labels is asymptotically the same. The local tree-like structure is preserved for finite $T$ (and $n \to \infty$) but the boundary conditions imposed by $t = 1$ and $t = T$ must be accounted for.

This said, in [29], the authors show that, for a Galton-Watson tree in which only spatial children are present, label reconstruction is feasible if and only if $c\Phi\lambda^2 = \alpha^2 > 1 = \alpha_c^2$, where $\lambda = (c_{\text{in}} - c_{\text{out}})/(c_{\text{in}} + c_{\text{out}})$. In [14], the authors conjectured a generalization of this result for a multi-type branching process, such as just described to construct $\mathcal{T}(v)$. In this setting, each node acts differently depending on its being a spatial or a temporal child. In the former case, two temporal children are generated (with label covariance equal to $\eta$), while in the latter only one temporal child is generated. The conjecture of [14] (which we adapted to the D-DCSBM) states that, for $T \to \infty$, community detection is possible if and only if the largest eigenvalue of

$$M_\infty(\alpha, \eta) = \begin{pmatrix} \alpha^2 & 2\eta^2 \\ \alpha^2 & \eta^2 \end{pmatrix} \tag{10}$$

is greater than one. This condition is verified as long as $\alpha > \alpha_c(\infty, \eta) = \sqrt{(1 - \eta^2)/(1 + \eta^2)}$.

The authors of [14] also provided directions to extend their result to finite $T$, which we here make explicit. For each time instant, three types of edges exist: spatial edges (connecting nodes in $\mathcal{G}_t$ to nodes in $\mathcal{G}_t$), forward temporal edges (connecting nodes in $\mathcal{G}_t$ to nodes in $\mathcal{G}_{t+1}$) and backwards temporal edges (connecting nodes in $\mathcal{G}_t$ to nodes in $\mathcal{G}_{t-1}$). We then construct a matrix $\tilde{M}_T(\alpha, \eta) \in \mathbb{R}^{3T \times 3T}$ identifying the rows and the columns as $\{(\text{backwards temporal})_t, (\text{spatial})_t, (\text{forward temporal})_t\}_{t=1,\dots,T}$. A $(\text{backwards temporal})_t$ edge goes from a node in $\mathcal{V}_t$ to a node in $\mathcal{V}_{t-1}$ that has, on average, $c\Phi$ spatial children with label correlation equal to $\lambda$ and one backwards temporal child, with label correlation equal to $\eta$. Similarly $(\text{spatial})_t$ goes from a node in $\mathcal{V}_t$ to a node in $\mathcal{V}_t$ having $c\Phi$ temporal children and, one forward and one backwards temporal children; finally, $(\text{forward temporal})_t$ goes from $\mathcal{V}_t$ to $\mathcal{V}_{t+1}$

Figure 5: Position of $\alpha_c(T, \eta)$ as a function of $\eta$ (x axis) and $T$ (color code). The black and red solid lines correspond to the analytical values of $\alpha_c(T, \eta)$, for $T = 1$ and $T = \infty$, respectively. The dashed dotted lines are the position of $\alpha_c(T, \eta)$ computed numerically, and the thick solid pale lines are the analytical values of $\alpha_c(T, \eta)$ for $T \in \{2, 3, 4\}$.

with one forward temporal child and $c\Phi$ spatial children. The entry $i, j$ of $\tilde{M}_T(\alpha, \eta)$ is then set equal to the number of off-springs of type $j$ of a node reached by an edge of type $i$, multiplied by the square label correlation. As forward temporal edges do not exist for $t = T$ and backwards temporal edges do not exist for $t = 1$, the matrix $\tilde{M}_T(\alpha, \eta) \in \mathbb{R}^{3T \times 3T}$ takes the form

$$\tilde{M}_T(\alpha, \eta) = \begin{pmatrix} \tilde{M}_d^+ & M_+ & 0 & \dots & 0 \\ \tilde{M}_- & M_d & \ddots & \dots & 0 \\ 0 & M_- & \ddots & M_+ & 0 \\ \vdots & \vdots & \ddots & M_d & \tilde{M}_+ \\ 0 & 0 & \dots & M_- & \tilde{M}_d^- \end{pmatrix} \tag{11}$$

where

$$M_d = \begin{pmatrix} 0 & 0 & 0 \\ \eta^2 & c\Phi\lambda^2 & \eta^2 \\ 0 & 0 & 0 \end{pmatrix}; \quad M_+ = \begin{pmatrix} 0 & 0 & 0 \\ 0 & 0 & 0 \\ 0 & c\Phi\lambda^2 & \eta^2 \end{pmatrix}; \quad M_- = \begin{pmatrix} \eta^2 & c\Phi\lambda^2 & 0 \\ 0 & 0 & 0 \\ 0 & 0 & 0 \end{pmatrix}$$

$$\tilde{M}_d^+ = \begin{pmatrix} 0 & 0 & 0 \\ 0 & c\Phi\lambda^2 & \eta^2 \\ 0 & 0 & 0 \end{pmatrix}; \quad \tilde{M}_+ = \begin{pmatrix} 0 & 0 & 0 \\ 0 & 0 & 0 \\ 0 & c\Phi\lambda^2 & 0 \end{pmatrix};$$

$$\tilde{M}_- = \begin{pmatrix} 0 & c\Phi\lambda^2 & 0 \\ 0 & 0 & 0 \\ 0 & 0 & 0 \end{pmatrix}; \quad \tilde{M}_d^- = \begin{pmatrix} 0 & 0 & 0 \\ \eta^2 & c\Phi\lambda^2 & 0 \\ 0 & 0 & 0 \end{pmatrix}.$$

Note that, since the first and the last rows of $\tilde{M}_T(\alpha, \eta)$ only have zero entries, $\tilde{M}_T(\alpha, \eta)$ has the same non-zero eigenvalues as $M_T(\alpha, \eta)$ defined in Equation (3). This also implies that $M_T(\alpha, \eta)$ shares the non-zero eigenvalues of a matrix of size $(3T - 2) \times (3T - 2)$ as initially conjectured in [14].

The analytical expression of $\alpha_c(T, \eta)$ can be obtained for $T = 2, 3, 4$ and is reported in the main text. For all other values of $T$ it can be computed numerically. The value of $\alpha_c(T, \eta)$ as a function of $\eta$ is reported in Figure 5 for different values of $T$.

## B Derivation of the dynamic Bethe-Hessian matrix

This appendix derives the matrix $H_{\xi,h}$, which arises from the variational Bethe approximation applied to the Hamiltonian of Equation (6), which we recall assumes the form

$$\mathcal{H}_{\xi,h}(s) = -\sum_{t=1}^{T}\left(\sum_{(i_t,j_t)\in\mathcal{E}_t}\mathrm{ath}(\xi)s_{i_t}s_{j_t} + \sum_{i_t\in\mathcal{V}_t}\mathrm{ath}(h)s_{i_t}s_{i_{t+1}}\right). \tag{12}$$

Collecting all time instants, $\mathcal{H}_{\xi,h}(s)$ can be synthetically written under the form

$$\mathcal{H}_{\xi,h}(s) = -\sum_{(ij)\in\mathcal{E}}\mathrm{ath}(\omega_{ij})\,s_is_j \tag{13}$$

for some appropriate coupling $\omega_{ij}$ (and where we recall that $\mathcal{E}$ is the set of all edges of $\mathcal{G}$). Each realization $s$ is a random variable, drawn from the Maxwell-Boltzmann distribution

$$\mu(s) = \frac{1}{Z}e^{-\mathcal{H}_{\xi,h}(s)}, \tag{14}$$

where $Z$ is the normalization constant. We are interested in the average realization of $s$ over the distribution $\mu(\cdot)$, that we denote $m^* = \langle s\rangle$, with $\langle\cdot\rangle$ being the average over (14). From Equation (14), configurations having a small energetic cost will occur with a larger probability but there are very few such configurations, as opposed to the exponentially many disordered ones. The competing behavior of these two terms defines two regimes: (i) the small interaction regime (called the *paramagnetic phase*, for small $\xi$ and $h$) in which the disordered configurations dominate the average configuration (which is the null vector) and (ii) the strong interaction regime (for large $\xi$ and $h$) in which the average value of $s$ is non-trivial and is dominated by the *modes* of $s$ which are local minima of the Hamiltonian of Equation (6). These modes are determined by the "mesoscale" structure of $\mathcal{G}$.

The value of $m^*$ cannot be computed exactly but, given the locally tree-like nature of $\mathcal{G}$, it may be evaluated using the asymptotically exact variational Bethe approximation [47]. This approximation $P_{\mathrm{Bethe}}(\cdot)$ of $\mu(\cdot)$ reads

$$P_{\mathrm{Bethe}}(s) = \frac{\prod_{(i,j)\in\mathcal{E}}P_{ij}(s_is_j)}{\prod_{i\in\mathcal{V}}[P_i(s_i)]^{d_i-1}}, \tag{15}$$

where $P_{ij}(\cdot)$ and $P_i(\cdot)$ are the edge and node marginals of $P_{\mathrm{Bethe}}$ and $d_i$ is the *total degree on $\mathcal{G}$* of node $i$. Further defining the free energy and the Bethe free energy respectively as

$$F = \sum_s \mu(s)\left[\mathcal{H}_{\xi,h}(s) + \log\mu(s)\right] = -\log Z \tag{16}$$

$$F_{\mathrm{Bethe}}(m,\chi) = \sum_s P_{\mathrm{Bethe}}(s)\left[\mathcal{H}_{\xi,h}(s) + \log P_{\mathrm{Bethe}}(s)\right], \tag{17}$$

where $m_i = \langle\sigma_i\rangle_{\mathrm{Bethe}}$ and $\chi_{ij} = \langle\sigma_i\sigma_j\rangle_{\mathrm{Bethe}}$, $\langle\cdot\rangle_{\mathrm{Bethe}}$ denoting the average taken over $P_{\mathrm{Bethe}}(\cdot)$. From a direct calculation, it comes that $F_{\mathrm{Bethe}}(m,\chi) - F = D_{\mathrm{KL}}(P_{\mathrm{Bethe}}||\mu) \geq 0$, where $D_{\mathrm{KL}}(\cdot)$ is the Kullback-Leibler divergence. Therefore, by minimizing $F_{\mathrm{Bethe}}$ with respect to $m$, one minimizes the divergence with respect to the real distribution and obtains an optimal estimate for $m^*$.

The Bethe free energy can be obtained by plugging Equation (15) into Equation (17) and takes the explicit form

$$F_{\mathrm{Bethe}}(m,\chi) = -\sum_{(ij)\in\mathcal{E}}\mathrm{ath}(\omega_{ij})\,\chi_{ij} + \sum_{(ij)\in\mathcal{E}}\sum_{s_is_j}f\left(\frac{1+m_is_i+m_js_j+\chi_{ij}s_is_j}{4}\right)$$
$$-\sum_{i\in\mathcal{V}}(d_i-1)\sum_{s_i}f\left(\frac{1+m_is_i}{2}\right), \tag{18}$$

where $f(x) = x\log(x)$. In the case of weak interactions (small $\omega_{ij}$), $F_{\mathrm{Bethe}}$ has a unique minimum in $m = 0$. For larger values of $\omega_{ij}$, it has a global minimum at $m \propto \mathbf{1}_{nT}$ and other local minima appear, corresponding to configurations correlated with the mesoscale structure of $\mathcal{G}$. In order to

study along which directions the function $F_{\text{Bethe}}$ finds its local minima, one needs to evaluate the Hessian matrix of $F_{\text{Bethe}}$ at $\boldsymbol{m} = 0$, as done in [32, 19], to obtain

$$\left.\frac{\partial^2 F_{\text{Bethe}}(\boldsymbol{m}, \boldsymbol{\chi})}{\partial m_i \partial m_j}\right|_{\boldsymbol{m}=0} = -\frac{\chi_{ij}}{1 - \chi_{ij}^2} A_{ij} + \left(\sum_{k \in \partial i} \frac{1}{1 - \chi_{ik}^2} - (d_i - 1)\right) \mathbb{1}_{ij}, \qquad (19)$$

where $A \in \{0, 1\}^{nT}$ is the adjacency matrix of $\mathcal{G}$ and $d_i = [A\mathbf{1}_n]_i$. Similarly minimizing $F_{\text{Bethe}}$ with respect to $\chi_{ij}$,

$$\left.\frac{\partial F_{\text{Bethe}}(\boldsymbol{m}, \boldsymbol{\chi})}{\partial \chi_{ij}}\right|_{\boldsymbol{m}=0} = -\text{ath}(\omega_{ij}) + \text{ath}(\chi_{ij}) = 0 \qquad (20)$$

and so $\chi_{ij} = \omega_{ij}$.

To finally retrieve the expression of Equation (7), note that $d_i = d_i^{(t)} + 2$ if $2 \leq t \leq T - 1$ and $d_i = d_i^{(t)} + 1$ otherwise, where $d_i^{(t)}$ is the degree of node $i$ in $\mathcal{G}_t$, and impose

$$\chi_{ij} = \begin{cases} \xi & \text{if } \exists\, t \text{ such that } i, j \in \mathcal{V}_t \\ h & \text{otherwise} \end{cases} \qquad (21)$$

as requested.

We therefore retrieve the matrix $H_{\xi,h}$ of Equation (7). When $H_{\xi,h}$ has a negative eigenvalue, $\boldsymbol{m} = 0$ is a saddle point and the free energy has a local minimum for some non-trivial configuration. The eigenvector associated to this negative eigenvalue points towards the direction of the local minimum of $F_{\text{Bethe}}$. As discussed in Section 3, the directions along which stable configurations are observed correspond to the dominant modes appearing in the Hamiltonian and are naturally correlated to the class structure. The smallest eigenvalue-eigenvector pairs of $H_{\xi,h}$ may thus be used to retrieve information on the directions of the dominant informative modes of the graph, as depicted in Figure 2.

## C   Technical results of Section 3.2

This section provides theoretical support to Proposition 1 of Section 3.2.

Exploiting the deep relation – which we detail in Section C.1 – that there is between the dynamical Bethe-Hessian of Equation (7) and the weighted non-backtracking matrix of Equation (8), we study the spectrum of the latter to infer some important properties of our proposed dynamical Bethe-Hessian. In particular, the eigenvalues of the non-backtracking matrix can be divided into two groups: (i) a majority of eigenvalues contained in a disc in the complex plane which delimits the *bulk* of this matrix (ii) few isolated eigenvalues with modulus larger than the radius of the bulk. These properties are known and well established in the static regime [20, 12] and we empirically observed to be maintained also in the dynamical setting under study. Furthermore, in the case of $k = 2$ classes, in the static case, the isolated eigenvectors (with largest modulus) are the Perron-Frobenius eigenvector (with all positive entries) and the eigenvector useful for community reconstruction. Similarly, in the dynamical case we have two *families* of eigenvectors (see Figure 2) coming from these two modes. We will refer to them as *informative family* and *uninformative family*.

Based on these empirical observations, we formulate the following assumption:

**Assumption 1** *Let $\mathcal{G}$ be a graph generated according to Definition 1 and $B_{\xi,h}$ the matrix defined in Equation (8). The bulk of $B_{\xi,h}$ is bounded by a disk in the complex plane with radius denoted by $L_{\xi,h}$. The eigenvalues with modulus larger than $L_{\xi,h}$ are isolated and their corresponding eigenvector are determined by the mesoscale structure of $\mathcal{G}$.*

Based on this assumption, in Section C.2 we determine the asymptotic position of the isolated eigenvalues with modulus larger than the radius of the bulk, as well as the radius of the bulk itself; from these results, Sections C.3 concludes on Proposition 1. In passing, with the results of C.2, some properties of the spectrum of the dynamical non-backtracking matrix of [14] are also discussed.

## C.1 Bethe-Hessian and weighted non-backtracking matrices

Let us first elaborate on an important property connecting the spectra of the Bethe-Hessian and non-backtracking matrices. This relation is well known in the literature (see *e.g* [48, 31, 32]). For sake of clarity, we here report the main results that relate the eigenvalues and eigenvectors of the two matrices. Let us consider the following two matrices for arbitrary weights $\boldsymbol{\omega} = \{\omega_{ij}\}_{(ij)\in\mathcal{E}}$ such that $\omega_{ij} < 1$ for all $(ij) \in \mathcal{E}^d$, the set of directed edges of $\mathcal{G}$:

$$(B_{\boldsymbol{\omega}})_{(ij)(k\ell)} = \mathbb{1}_{jk}(1 - \mathbb{1}_{il})\,\omega_{kl} \quad \forall\,(ij),(kl) \in \mathcal{E}^d, \tag{22}$$

$$(H_{\boldsymbol{\omega}})_{ij} = \left(1 + \sum_{k\in\partial i}\frac{(\omega_{ik}/x)^2}{1 - (\omega_{ik}/x)^2}\right)\mathbb{1}_{ij} - \frac{(\omega_{ik}/x)}{1 - (\omega_{ik}/x)^2}A_{ij}, \quad \forall\,i,j\in\mathcal{V},\ x\in(1,\infty) \tag{23}$$

We now show that, for $\omega_{ij} < 1$, whenever $x \geq 1$ is a real eigenvalue of $B_{\boldsymbol{\omega}}$, $\det[H_{\boldsymbol{\omega}/x}] = 0$. Indeed, let $\boldsymbol{g} \in \mathbb{R}^{|\mathcal{E}^d|}$ be an eigenvector of $B_{\boldsymbol{\omega}}$ with eigenvalue $x \geq 1$. Then

$$(B_{\boldsymbol{\omega}}\boldsymbol{g})_{ij} = \sum_{k\in\partial j\setminus i}\omega_{jk}g_{jk} = m_j - \omega_{ji}g_{ji} = xg_{ij}, \tag{24}$$

where $m_j \equiv \sum_{k\in\partial j}\omega_{jk}g_{jk}$. We may gather this relation under the system of equations

$$\begin{pmatrix} m_j \\ m_i \end{pmatrix} = \begin{pmatrix} x & \omega_{ij} \\ \omega_{ij} & x \end{pmatrix}\begin{pmatrix} g_{ij} \\ g_{ji} \end{pmatrix}. \tag{25}$$

Since $\omega_{ij}^2 < 1$ for all $(i,j)$, the system is invertible and a straightforward calculation gives

$$m_i = \sum_{j\in\partial i}\frac{\omega_{ij}x}{x^2 - \omega_{ij}^2}m_j - m_i\sum_{j\in\partial i}\frac{\omega_{ij}^2}{x^2 - \omega_{ij}^2} \tag{26}$$

which eventually leads to

$$H_{\boldsymbol{\omega}/x}\boldsymbol{m} = 0. \tag{27}$$

This confirms that, not only there is a connection among the spectra of the Bethe-Hessian and non-backtracking matrices, but also between their eigenvectors. Note that, by choosing $\omega_{ij} = \xi$ is there if $t$ such that $i,j \in \mathcal{V}_t$ and $\omega_{ij} = h$ otherwise, we precisely recover the definitions of $B_{\xi,h}$ and $H_{\xi/x,h/x}$ as per Equations (7, 8).

We now further comment how the spectra of $B_{\xi,h}$ and $H_{\xi/y,h/y}$ are related when $y \in \mathbb{R}$ is not an eigenvalue of $B_{\xi,h}$. First recall that, as per Assumption 1, the large majority of the eigenvalues of $B_{\xi,h}$ are asymptotically bounded by a circle in the complex plane and that only few isolated eigenvalues are larger in modulus with associated eigenvectors representative of the mesoscale structure of $\mathcal{G}$. First consider the case where $y \to \infty$. Then, letting $\tilde{\xi} = \xi/y \to 0$ and $\tilde{h} = h/y \to 0$, by definition (Equation (7)), it comes that $H_{\tilde{\xi},\tilde{h}} \succ 0$, *i.e.*, all the eigenvalues are positive. Now, decreasing $y$ to $y = \rho(B_{\xi,h})$, from Equation (27), $H_{\tilde{\xi},\tilde{h}}$ has one eigenvalue equal to zero, which is necessarily the smallest and for all $y > \rho(B_{\xi,h})$, $H_{\tilde{\xi},\tilde{h}}$ is positive definite. This is because if there was a $y > \rho(B_{\xi,h})$ such that $\det[H_{\tilde{\xi},\tilde{h}}] = 0$, then $y$ would have to be an eigenvalue of $B_{\xi,h}$, which is absurd by construction.

For $y$ lying between the first and the second largest real eigenvalues of $B_{\xi,h}$, no eigenvalue of $H_{\tilde{\xi},\tilde{h}}$ is equal to zero, and the smallest one is negative and isolated. Further decreasing the value of $y$, the smallest (isolated) eigenvalues of $H_{\tilde{\xi},\tilde{h}}$ become progressively negative in correspondence of the largest isolated eigenvalues of $B_{\xi,h}$.

Formally, this discussion may be summarized as follows.

**Property 1** *Let $L_{\xi,h}$ be the radius of the bulk of $B_{\xi,h}$ and let $y \geq L_{\xi,h}$. Then, the number of real (isolated) eigenvalues of $B_{\xi,h}$ which are greater (or equal) to $y$ is equal to the number of (isolated) eigenvalues of $H_{\xi/y,h/y}$ which are smaller (or equal) to zero. In particular, for $y = L_{\xi,h}$, the left edge of bulk spectrum of $H_{\xi/y,h/y}$ is asymptotically close to $0^+$.*

A pictorial representation of Property 1 is given in Figure 6. With this result, we know how to relate the spectrum of $H_{\xi,h}$ to the spectrum of $B_{\xi,h}$ that we study in the next section.

Figure 6: **Left** : spectrum of the matrix $B_{\xi,h}$ in the complex plane. In blue the considered value of $y$ and in larger size the two eigenvalues of $B_{\xi,h}$ larger than $y$. **Right** : histogram of $H_{\xi/y,h/y}$ with evidenced the two negative eigenvalues. **For both simulations**: $n = 1\,000$, $T = 3$, $k = 4$, $c = 6$, $c_{\text{out}} = 2$ for all off-diagonal elements of $C$, $\Phi = 1$, $\eta = 0.9$, $\xi = 0.8$, $h = 0.6$ and $y = 4.2$.

## C.2 Spectrum of the weighted non-backtracking matrix

We now proceed in our agenda by studying the spectrum of $B_{\xi,h}$ under Assumption 1. The method we use can be seen as a generalization of [18]. By considering the expression of the expected eigenvector, we first determine the position of the eigenvalues belonging to the *informative family* (starting from the largest) and then of the *uninformative family*. Secondly, we analyze the variance of the expression of the expected eigenvector and see under what condition the expectation is meaningful. With this result we finally determine the value of $L_{\xi,h}$ (the radius of the bulk of $B_{\xi,h}$) and summarize our findings in Proposition 2.

### C.2.1 The position of the informative eigenvalues

In this section we determine the position of the informative eigenvalues of $B_{\xi,h}$ with modulus larger than $L_{\xi,h}$. To do so, we first study the largest of them in the limiting case $T \to \infty$, to then extend our findings for finite $T$ to all other eigenvalues.

**The limiting case of** $T \to \infty$

Consider the graph $\mathcal{G}$ generated according to Definition 1. Let $\omega_{ij} = \xi$ if there exists $t$ such that $i, j \in \mathcal{V}_t$ and $\omega_{ij} = h$ otherwise, and let $\boldsymbol{g}^{(r)} \in \mathbb{R}^{|\mathcal{E}|^d}$, for $r \in \mathbb{N}$, be the vector with entry

$$g_{ij}^{(r)} = \frac{1}{\mu_1^r} \sum_{\substack{(wx) \,:\, d(jk,wx)=r \\ k \neq i}} W_{(jk)\to(wx)}\sigma_x, \tag{28}$$

where $\{(jk) \;:\; d(jk,wx) = r\}$ is the set of directed edges $(jk)$ such that the shortest directed non-backtracking path connecting $(jk)$ to $(wx)$ is of length $r$, and where $W_{(jk)\to(wx)}$ is the "total weight" of this shortest path defined as the product of each edge weight $\omega_{ij}$, *i.e*,

$$W_{(jk)\to(wx)} = \omega_{(jk)}\omega_{(k\cdot)}\cdots\omega_{(\cdot w)}\omega_{(wx)}. \tag{29}$$

The quantity $\sigma_x \in \{\pm 1\}$ takes its value according to the label of node $x$. The value of $\mu_1$ appearing in Equation (28) will be chosen in order to enforce the vector $\boldsymbol{g}^{(r)}$ to be an approximate eigenvector of $B_{\xi,h}$, defined in Equation (22). By the definition of $\boldsymbol{g}^{(r)}$, recalling the expression of $B_{\xi,h}$ in (22), we find that

$$(B_{\xi,h}\boldsymbol{g}^{(r)})_{ij} = \mu_1 g_{ij}^{(r+1)}. \tag{30}$$

We now analyze this expression exploiting the tree-like approximation elaborated in Appendix A. Resuming from this approximation, the expectation of $g_{ij}^{(r)}$ may be written under the following form:

$$\mathbb{E}[g_{ij}^{(r)}] = \frac{1}{\mu_1^r} \left( c\Phi\lambda\xi\chi_s^{(r-1)} + \phi_i\eta h\chi_t^{(r-1)} \right) \sigma_j. \tag{31}$$

Here the first addend is the contribution of the spatial children of $j$ which are on average $c\Phi$ in number, and for each of them the weight of the connecting edge is equal to $\xi$ while the correlation between the labels $\lambda = \mathbb{E}[\sigma_j\sigma_k]$. Each spatial child being at a distance $r-1$ from the target edges – themselves at a distance $r$ from $(jk)$ – contributes to the sum through a term which we denoted $\chi_s^{(r-1)} > 0$. Similarly, the second addend is the contribution of the temporal children which are $\phi_i = 2$ in number if $(ij)$ is a spatial edge or $\phi_i = 1$ if $(ij)$ is a temporal edge; their own contribution is denoted $\chi_t^{(r-1)} > 0$. The correlation of the labels of temporal children is equal to $\eta$ and the weight of the edges is equal to $h$. Importantly note that, as a consequence of $\lambda, \xi, \eta, h$ being assumed to be all positive, both $\chi_s^{(r)}$ and $\chi_t^{(r)}$ are positive as well.

By recurrence, the values of $\chi_{s/t}^{(r)}$, which we just defined, then undergo the following relation

$$\begin{pmatrix} \chi_s^{(r)} \\ \chi_t^{(r)} \end{pmatrix} = \begin{pmatrix} c\Phi\lambda\xi & 2\eta h \\ c\Phi\lambda\xi & \eta h \end{pmatrix} \begin{pmatrix} \chi_s^{(r-1)} \\ \chi_t^{(r-1)} \end{pmatrix} = \begin{pmatrix} c\Phi\lambda\xi & 2\eta h \\ c\Phi\lambda\xi & \eta h \end{pmatrix}^r \begin{pmatrix} \chi_s^{(0)} \\ \chi_t^{(0)} \end{pmatrix} \tag{32}$$

$$\equiv \left( M_\infty(\sqrt{c\Phi\lambda\xi}, \sqrt{h\eta}) \right)^r \begin{pmatrix} \chi_s^{(0)} \\ \chi_t^{(0)} \end{pmatrix}, \tag{33}$$

where $M_\infty(\cdot, \cdot)$ is the matrix introduced in Equation (3). For simplicity we will denote it as $M_\infty$. For, say, $r \sim \log(n)$, $\chi_{s/t}^{(r)} \approx \rho^r(M_\infty)v_{s/t}$, where $\boldsymbol{v} = (v_s, v_t)$ is the eigenvector associated to the eigenvalue of $M_\infty$ of largest amplitude. Equation (31) can therefore be further approximated as

$$\mathbb{E}[g_{ij}^{(r)}] = \left( \frac{\rho(M_\infty)}{\mu_1} \right)^r (c\Phi\lambda\xi v_s + \phi_i\eta h v_t)\,\sigma_j + o\left( \frac{\rho(M_\infty)}{\mu_1} \right)^r. \tag{34}$$

This expression naturally leads to the choice $\mu_1 = \rho(M_\infty)$ for which $\mathbb{E}[g_{ij}^{(r)}]$ is independent of $r$, thus turning Equation (30) into an approximate eigenvector equation and $\mu_1$ into a close approximation of one of the real eigenvalues of $B_{\xi,h}$.

We now extend this result to the case of finite $T$, and bring further conclusion on all the eigenvalues of $B_{\xi,h}$ belonging to the *informative family*.

**The case of finite $T$**

As we discussed already along Appendix A, the case of finite $T$ introduces further difficulties due to the time-boundaries $t = 1$ and $t = T$. This being accounted for, when analyzing the contribution of each edge, not only we have to distinguish between spatial and temporal edges, but also to specify the time at which the edge lives. More precisely, suppose that $j \in \mathcal{V}_t$ for $1 \leq t \leq T$. We can rewrite Equation (31) as

$$\mathbb{E}[g_{ij}^{(r)}] = \frac{1}{\mu_1} \left[ c\Phi\lambda\xi\chi_{s,t}^{(r-1)} + (1-\delta_{1,t})\eta h\chi_{b,t}^{(r-1)} + (1-\delta_{T,t})\eta h\chi_{f,t}^{(r-1)} \right], \tag{35}$$

where $\chi_{s,t}^{(\cdot)}, \chi_{b,t}^{(\cdot)}, \chi_{f,t}^{(\cdot)}$ are respectively the contributions to the of a spatial, a backwards temporal and a forward temporal child of a node $j \in \mathcal{V}_t$. The relation between all the $\chi$'s directly unfolds from the branching process at finite $T$ that we already discussed in Appendix A. More precisely, let $\boldsymbol{\chi}^{(r)} = \{\chi_{b,t}^{(r)}, \chi_{s,t}^{(r)}, \chi_{f,t}^{(r)}\}_{t=1,\dots,T}$, then the following relation holds:

$$\boldsymbol{\chi}^{(r)} = M_T\left( \sqrt{c\Phi\lambda\xi}, \sqrt{\eta h} \right) \boldsymbol{\chi}^{(r-1)}, \tag{36}$$

where $M_T(\cdot, \cdot)$ is the matrix defined in Equation (4). Following the argument we just detailed for $T \to \infty$, we then get that the largest eigenvalue of the *informative family* is asymptotically close to $\mu_1 = \rho\left( M_T\left( \sqrt{c\Phi\lambda\xi}, \sqrt{\eta h} \right) \right)$.

This analysis also allows us to describe the subsequent eigenvalues $\mu_{i\geq 2}$ belonging to the *informative family* that have a smaller modulus. These modes are metastable configurations of the branching process as in configuration 4 of Figure 2. In these modes, nodes belonging to different communities are still distinguished (hence the reason why these modes are *informative*), but the class identification $\sigma_x$ may be reversed across time. This results in a state in which neighbours are more likely to change

label than to keep it, hence they have *negative* label correlation and lead to negative values of $\chi$. This means to relax the constraint $\chi > 0$ and thus no longer looking for the leading eigenvalue of $M_T$. From this intuition we argue that the subsequent informative eigenvalues of $B_{\xi,h}$ coincide with the subsequent eigenvalues of $M_T \left( \sqrt{c\Phi\lambda\xi}, \sqrt{\eta h} \right)$.

A further important remark should be made on the eigenvalues $\mu_{i \geq 1}$. The matrix $M_T$ is real and non-negative, but it is not symmetric. Consequently, the leading eigenvalue, $\mu_1$ will certainly be real (due to Perron-Frobenius theorem), while the subsequent eigenvalues are potentially complex. Although we cannot offer a clear interpretation for the complex nature of some of these isolated eigenvalues, our study is experimentally verified to hold also in this case as shown in Figure 7.

We now proceed extending our arguments to the *uninformative family* of isolated eigenvalues of $B_{\xi,h}$.

### C.2.2   The position of the uninformative isolated eigenvalues

As in the static case, not all stable configurations of the branching process of Appendix A are informative. In particular, two nodes of $\mathcal{G}$ might be considered to belong to the same community only because they live at the same time. Based on the technique detailed in Section C.2.1, we now describe the position of the eigenvalues forming the *uninformative family*. Although these eigenvalues are not informative, the awareness of their presence is crucial if one has to avoid to mistakenly use one of these for community reconstruction.

We proceed again by studying the largest of these eigenvalues (which is also the largest eigenvalue of $B_{\xi,h}$), to then extended the result to all the others. Let us denote $\{\gamma_i\}_{i=1,...,T}$ this second set of (trivial and non-informative) eigenvalues. The approximate Perron-Frobenius eigenvector $\boldsymbol{b} \in \mathbb{R}^{2|\mathcal{E}|}$ can be written as

$$b_{ij}^{(r)} = \frac{1}{\gamma_1^r} \sum_{\substack{(wx) \,:\, d(jk,wx)=r \\ k \neq i}} W_{(jk)\to(wx)}. \tag{37}$$

According to this expression, we set $\sigma_x = 1$ for all nodes and thus the correlation between $\sigma_x$ and $\sigma_y$ is always unitary. Following the argument developed to determine the value of $\mu_1$, we then obtain

$$\gamma_1 = \rho \left( M_T \left( \sqrt{c\Phi\xi}, \sqrt{h} \right) \right). \tag{38}$$

As in Section C.2.1, this eigenvalue is necessarily real and the subsequent eigenvalues of the *uninformative family* are given by the subsequent eigenvalues of $M_T \left( \sqrt{c\Phi\xi}, \sqrt{h} \right)$ and can be complex. Note importantly that the ordering of $\{\mu_i\}_{i \geq 1}$ and $\{\gamma_i\}_{i \geq 1}$ is not *a priori* well defined.

So far we determined the position of the isolated eigenvalues under the assumption that the expectation of the approximate eigenvectors are significant. In order to know when this analysis holds, we have to study the variance of the entries of the approximate eigenvectors and see under what conditions it vanishes. This analysis will also allow us to determine the value of the radius of the bulk of $B_{\xi,h}$.

### C.2.3   The bulk eigenvalues of $B_{\xi,h}$

To begin with, we investigate under which conditions the approximate eigenvector Equations (31, 37) hold. We then proceed with a study of the variance of $g_{ij}^{(r)}$ (and $b_{ij}^{(r)}$). When the variance vanishes, the eigenvector is well approximated by its expectation and we conjecture it is isolated. On the contrary, when the variance diverges it is because it gets asymptotically close to the bulk of uninformative eigenvalues and is no longer isolated.

Let us first consider the eigenvector attached to $\mu_1$:

$$\mathbb{E}\left[ \left( g_{ij}^{(r)} \right)^2 \right] = \frac{1}{\mu_1^{2r}} \sum_{\substack{(wx) \,:\, d(jk,wx)=r \\ k \neq i}} \left( W_{(jk)\to(wx)}^2 + \sum_{\substack{(vy) \,:\, d(jk,vy)=r \\ (vy) \neq (wx), k \neq i}} \sigma_x \sigma_y W_{(jk)\to(wx)} W_{(jk)\to(vy)} \right). \tag{39}$$

The first addend of (39) can be evaluated as previously done in Equation (33), getting

$$\mathbb{E}\left[\frac{1}{\mu_1^{2r}} \sum_{\substack{(wx) \,:\, d(jl,wx)=r \\ l \neq i}} W_{(jl)\to(wx)}^2\right] = O\left(\frac{\rho^r\left(M_T(\sqrt{c\Phi\xi^2}, h)\right)}{\mu_1^{2r}}\right). \tag{40}$$

If $\mu_1^2 < \rho\left(M_T(\sqrt{c\Phi\xi^2}, h)\right)$, this addend of (39) diverges, and so does the variance of $g_{ij}^{(r)}$: in this case, $\boldsymbol{g}^{(r)}$ cannot be an approximate eigenvector of $B_{\xi,h}$.

Consider next the second addend of Equation (39):

$$\mathbb{E}\left[\frac{1}{\mu_1^{2r}} \sum_{\substack{(wx) \,:\, d(jk,wx)=r \\ l \neq i}} \sum_{\substack{(vy) \,:\, d(jk,vy)=r \\ (vy) \neq (wx), k \neq i}} \sigma_x \sigma_y W_{(jk)\to(wx)} W_{(jk)\to(vy)}\right]$$

$$= \frac{1}{\mu_1^{2r}} \sum_{\substack{(wx) \,:\, d(jk,wx)=r \\ l \neq i}} \sum_{\substack{(vy) \,:\, d(jk,vy)=r \\ (vy) \neq (wx), k \neq i}} \mathbb{E}[\sigma_j \sigma_x W_{(jk)\to(wx)} \cdot \sigma_j \sigma_y W_{(jk)\to(vy)}]$$

$$\approx \frac{1}{\mu_1^{2r}} \sum_{\substack{(wx) \,:\, d(jk,wx)=r \\ l \neq i}} \mathbb{E}[\sigma_j \sigma_x W_{(jk)\to(wx)}] \sum_{\substack{(vy) \,:\, d(jk,vy)=r \\ (vy) \neq (wx), k \neq i}} \mathbb{E}[\sigma_j \sigma_y W_{(jk)\to(vy)}]$$

$$\approx \frac{1}{\mu_1^{2r}} \sum_{\substack{(wx) \,:\, d(jk,wx)=r \\ l \neq i}} \mathbb{E}[\sigma_j \sigma_x W_{(jk)\to(wx)}] \sum_{\substack{(vy) \,:\, d(jk,vy)=r \\ k \neq i}} \mathbb{E}[\sigma_j \sigma_y W_{(jk)\to(vy)}]$$

$$= \mathbb{E}^2\left[g_{ij}^{(r)}\right], \tag{41}$$

where we exploited the fact that the paths $(jk \to wl)$ and $(jk \to vl)$ are asymptotically independent and that the number of paths leading to nodes a distance $r$ from $(jk)$ is exponentially large in $r$, unlike the number of paths leading to $(vy)$ from $(jk)$. We thus obtain that the variance $\mathbb{V}[g_{ij}^{(r)}]$ of $g_{ij}^{(r)}$ grows as

$$\mathbb{V}\left[g_{ij}^{(r)}\right] = O\left(\frac{\rho^r\left(M_T(\sqrt{c\Phi\xi^2}, h)\right)}{\mu_1^{2r}}\right). \tag{42}$$

As a consequence, the variance of $g_{ij}^{(r)}$ vanishes if and only if $\mu_1 > \sqrt{\rho\left(M_T(\sqrt{c\Phi\xi^2}, h)\right)}$.

Considering now the problem of evaluating the variance for all the $\{\mu_i\}_{i\geq 1}$ and $\{\gamma_i\}_{i\geq 1}$, note that, the variance is only determined by the first addend of Equation (39). This term does not depend on the configuration $\boldsymbol{\sigma}$ and is, therefore, the same for *all* the isolated eigenvectors. Consequently, for all the isolated eigenvectors, the variance vanishes if the corresponding eigenvalue is greater than $L_{\xi,h} = \sqrt{\rho\left(M_T(\sqrt{c\Phi\xi^2}, h)\right)}$, which is precisely the radius of the bulk of $B_{\xi,h}$, since an informative eigenvalue-eigenvector pair $(\mu_i, \boldsymbol{g}_i)$, (resp. $(\gamma_i, \boldsymbol{b}_i)$), for $B_{\xi,h}$ can only exist provided that $\mu_i$ (resp. $\gamma_i$) is greater than $L_{\xi,h}$.

The results of this section may be summarized as follows.

**Proposition 2** *Letting $\mathcal{G}$ be a graph generated as per Definition 1, in the $n \to \infty$ limit, the complex eigenvalues forming the bulk of $B_{\xi,h}$ are bounded by a disk in the complex plane of radius $L_{\xi,h} = \sqrt{\rho(M_T(\sqrt{c\Phi\xi^2}, h))}$, for $M_T(\cdot, \cdot)$ defined in Equation (4). All the eigenvalues of $B_{\xi,h}$ of magnitude larger than $L_{\xi,h}$ are isolated and are asymptotically close to one of the eigenvalues of either $M_T(\sqrt{c\Phi\xi\lambda}, \sqrt{\eta h})$ (in which case they correspond to non-trivial modes) or $M_T(\sqrt{c\Phi\xi}, \sqrt{h})$ (in which case they correspond to trivial modes).*

Figure 7: The 150 eigenvalues of $B_{\xi,h}$ with largest real part, for $n = 10\,000$, $T = 5$, $\eta = 0.4$, $c = 10$, $c_{\text{out}} = 4$, $\Phi = 1.64$. **Left** $\xi = 0.2$, $h = 0.9$. **Right** $\xi = 0.4$, $h = 0.7$. The blue dashed lines are the theoretical positions of the eigenvalues forming the *informative family*, while the black dashed-dotted lines indicate the *uninformative family*. The thickest blue and black lines are $\mu_1$ and $\gamma_1$, respectively. The imaginary eigenvalues are represented with a circle in the complex plane. The solid black line is a part of a circle of radius $L_{\xi,h}$.

Figure 7 confirms numerically Proposition 2 for two choices of values of $(\xi, h)$, one in which all the isolated eigenvalues are real and one in which there are complex isolated eigenvalues. We choose to compute only the 150 eigenvalues with largest real part to keep a reasonable computational time, while having a large value of $n$.

Based on these results, we now proceed giving the supporting arguments of Proposition 1.

## C.3 Supporting arguments for Proposition 1

This section provides the final theoretical support to Proposition 1 at the core of the article, being at the root of our proposed dynamic clustering algorithm. To this end, we need to show how the bulk spectrum of $B_{\xi,h}$ relates to the bulk spectrum of $H_{\xi,h}$ for the values of $(\xi, h)$ proposed in Proposition 1, *i.e.*, $\xi = \lambda_d = \frac{\alpha_c(T,\eta)}{\sqrt{c\Phi}}$ and $h = \eta$.

Exploiting the result of Proposition 2, the matrix $B_{\lambda_d,\eta}$ has an eigenvector correlated to the class labels equal to $\mu_1 = \rho(M_T(\sqrt{c\Phi\lambda\lambda_d}, \eta))$. First note that, by definition, $\sqrt{c\Phi\lambda_d^2} = \alpha_c(T,\eta)$, while $\sqrt{c\Phi\lambda^2} = \alpha$. For $\alpha > \alpha_c(T,\eta)$, then $\lambda > \lambda_d$, and, consequently $\sqrt{c\Phi\lambda\lambda_d} > \alpha_c(T,\eta)$. From this last equation and the definition of $\alpha_c(T,\eta)$ provided in Section 2.2, we conclude that $\mu_1 > 1$.

From Proposition 2, we further have that the radius of the bulk spectrum of $B_{\lambda_d,\eta}$ is equal to $L_{\lambda_d,\eta} = 1$. As such, the informative eigenvalue $\mu_1$ of $B_{\lambda_d,\eta}$ exists as soon as $\alpha > \alpha_c(T,\eta)$.

From Property 1, the smallest eigenvalue of the bulk (*i.e.*, its left-edge) of $H_{\lambda_d,\eta}$ is asymptotically close to zero and all the eigenvectors associated to the negative eigenvalues are correlated to the mesoscale structure of $\mathcal{G}$, thereby entailing the validity and optimal performance down to the detectability threshold of our proposed Algorithm 1.

In Figure 8 (subplots 2 and 4) we provide numerical support to Proposition 1, showing the spectra of $B_{\lambda_d,\eta}$ and $H_{\lambda_d,\eta}$.

## C.4 Analysis of the spectrum of $B_{\lambda,\eta}$

In the previous sections we studied the spectrum of $B_{\xi,h}$ for generic $(\xi, h)$. We now focus on the particular choice $(\xi = \lambda, h = \eta)$ that leads to $B_{\lambda,\eta}$, sharing the same eigenvalues of the dynamical non-backtracking of [14]. First we show that this matrix has an informative isolated eigenvalue (not necessarily the second largest) for all $\alpha > \alpha_c(T,\eta)$. We then show that the matrix $H_{\lambda,\eta}$ shares the same property. We further comment that, however, the choice $(\xi = \lambda, \eta = h)$ is impractical from an algorithmic standpoint.

Figure 8: **Sub-figures 1, 2**: spectrum of $B_{\xi,\eta}$ for $\xi = \lambda$ and $\xi = \lambda_d$, respectively. The green dashed line is the position of 1, while the black circle is of radius $L_{\xi,\eta}$. **Sub-figures 3, 4**: histogram of $H_{\xi,\eta}$ for $\xi = \lambda$ and $\xi = \lambda_d$, respectively. The black dashed line indicates the position of 0. For all simulations, $T = 2$, $\eta = 0.4$, $c = 6$, $c_{\text{out}} = 1$, $\Phi = 1$, $n = 2\,000$.

**Community detectability with $B_{\lambda,\eta}$**

The fact that the matrix $B_{\lambda,\eta}$ can be used for community reconstruction is a straightforward consequence of Proposition 2. In fact, letting $\xi = \lambda$ and $h = \eta$, we obtain that the leading informative eigenvalue is equal to $\mu_1 = \rho(M_T(\alpha,\eta))$, while the radius of the bulk is equal to $L_{\lambda,\eta} = \sqrt{\rho(M_T(\alpha,\eta))} = \sqrt{\mu_1}$. By definition, if $\alpha > \alpha_c(T,\eta)$, then $\mu_1 > 1$, therefore $\mu_1 > L_{\lambda,\eta}$. So for all $\alpha > \alpha_c(T,\eta)$, $\mu_1$ is an isolated eigenvalue in the spectrum of $B_{\lambda,\eta}$, but it does not correspond, in general, to the second largest eigenvalue.

We now proceed our discussion studying the matrix $H_{\lambda,\eta}$.

**Community detectability with $H_{\lambda,\eta}$**

In order to fully grasp the properties of the matrix $H_{\lambda,\eta}$, one has to consider its relation with $B_{\lambda,\eta}$ and the *belief propagation* (BP) equations. Specifically this allows us to show that the most informative eigenvalue of $B_{\lambda,\eta}$ is $1 < L_{\lambda,\eta}$ and lies isolated inside the bulk. Consequently, as per Section C.1, the most informative eigenvalue of $H_{\lambda,\eta}$ is equal to zero.

We first establish that $B_{\lambda,\eta}$ naturally comes into play by linearizing BP equations: these consist of a set of fixed-point equations defining "messages" $m_{ij}$ exchanged between the nodes $i$ and $j$, and ultimately providing an asymptotically optimal community clustering algorithm. Specifically, from the expression of the whole set of messages $m_{ij}$, one can estimate the marginal probability distribution of the label of each node. To this end, first define

$$H = \begin{pmatrix} \frac{1+\eta}{2} & \frac{1-\eta}{2} \\ \frac{1+\eta}{2} & \frac{1-\eta}{2} \end{pmatrix}, \quad C = \begin{pmatrix} c_{\text{in}} & c_{\text{out}} \\ c_{\text{out}} & c_{\text{in}} \end{pmatrix}. \tag{43}$$

Letting $a, b \in \{\pm 1\}$, the BP equations take the form [14, Equations 5,6,8]

$$m_{j_t,i_t}(a) = \frac{e^{-h_t(a)}}{Z_{j_t,i_t}} \left( \sum_b H_{ab}\, m_{i_t,i_{t+1}}(b) \right) \left( \sum_b H_{ab}\, m_{i_t,i_{t-1}}(b) \right) \prod_{l_t \in \partial i_t \setminus j_t} \sum_b C_{ab} m_{i_t,l_t}(b)$$

$$m_{i_{t+1},i_t}(a) = \frac{e^{-h_t(a)}}{Z_{i_{t+1},i_t}} \left( \sum_b H_{ab}\, m_{i_t,i_{t-1}}(b) \right) \prod_{l_t \in \partial i_t} \sum_b C_{ab} m_{i_t,l_t}(b) \tag{44}$$

where

$$h_t(a) = \frac{1}{n} \sum_{j \in \mathcal{V}_t} \sum_b C_{ab} m_{i_t,j_t}(b). \tag{45}$$

The above messages can be expanded around the so-called *trivial* fixed point[9] $m_{j_t,i_t}(\pm 1) = 1/2 \pm \epsilon_{i_t,j_t}$, $m_{i_t,i_{t\pm 1}}(\pm 1) = 1/2 \pm \epsilon_{i_t,i_{t\pm 1}}$, yielding

$$\epsilon_{j_t,i_t} = \eta(\epsilon_{i_t,i_{t-1}} + \epsilon_{i_t,i_{t+1}}) + \lambda \sum_{\ell_t \in \partial i_t \setminus j_t} \epsilon_{i_t,\ell_t} \tag{46}$$

$$\epsilon_{i_{t+1},i_t} = \eta\epsilon_{i_t,i_{t-1}} + \lambda \sum_{\ell_t \in \partial i_t} \epsilon_{i_t,\ell_t}. \tag{47}$$

These equations can be rewritten in synthetic form introducing the weighted non-backtracking matrix

$$B_{\lambda,\eta}\boldsymbol{\epsilon} = \boldsymbol{\epsilon}. \tag{48}$$

In agreement with our empirical observations, we predict that the matrix $B_{\lambda,\eta}$ has an eigenvalue asymptotically close to one, so that, as a consequence of the property discussed in Appendix C.1, $H_{\lambda,\eta}$ has an eigenvalue asymptotically close to zero. The corresponding eigenvector of $B_{\lambda,\eta}$ represents the deviation from the trivial fixed point and is naturally connected to the community structure. The presence (and importance) of this isolated eigenvalue has been already observed and studied in the static regime [28, 49] and is visually depicted in Figure 8 (subplots 1 and 3).

We finally argue that this eigenvalue of $B_{\lambda,\eta}$ exists and is isolated as soon as $\alpha > \alpha_c(T,\eta)$. Indeed, the eigenvalue equal to one lies *isolated inside* the bulk of $B_{\lambda,\eta}$, the radius of the bulk spectrum of $B_{\lambda,\eta}$ being $L_{\lambda,\eta} = \sqrt{\rho(M_T(\alpha,\eta))}$. There further exists another informative eigenvalue which is equal to $\mu_1 = \rho(M_T(\alpha,\eta))$. The eigenvalue equal to 1 remains isolated inside the bulk for all $\alpha > \alpha_c(T,\eta)$ and meets the outer-bulk isolated eigenvalue, $\mu_1$, right at the edge of the bulk when $\alpha = \alpha_c(T,\eta)$ (*i.e.*, at the precise detection threshold). Below the transition threshold, when $\alpha < \alpha_c(T,\eta)$, the two eigenvalues then become complex conjugate.

This result can be summarized in the form of the following proposition.

**Proposition 3** *Let $\mathcal{G}$ be a graph generated as per Definition 1. As $n \to \infty$, the complex eigenvalues forming the bulk of of the non-symmetric matrix $B_{\lambda,\eta}$ are asymptotically bounded by a circle in the complex plane of radius $L_{\lambda,\eta} = \sqrt{\rho(M_T(\alpha,\eta))}$, with $\alpha = \sqrt{c\Phi\lambda^2}$ and $M_T(\alpha,\eta)$ defined in (3).*

*Besides, if $\alpha > \alpha_c(T,\eta)$, then $1 < L_{\lambda,\eta}$, 1 is an isolated eigenvalue of $B_{\lambda,\eta}$ and 0 is an isolated eigenvalue of $H_{\lambda,\eta}$, and the corresponding eigenvectors for both matrices are correlated to the vector of community labels.*

Proposition 3 states that one informative eigenvector of $H_{\lambda,\eta}$ (the one corresponding to the mode 2 of Figure 2) is associated to the zero eigenvalue, but nothing is said on its relative position in the spectrum of $H_{\lambda,\eta}$. This is a practical issue: indeed, as $\lambda$ is also a priori unknown, one cannot simply browse over values of $\lambda$ in search for an isolated zero eigenvalue of $H_{\lambda,\eta}$, which may correspond to a non-informative mode.

The numerical support of Proposition 3 (subplots 1 and 3) is provided by Figure 8.

# D  Dependence of the realizations of $A^{(t)}$ by adding edge persistence

This section provides hints to generalize the main results of the article to networks with persistence not only in the labels, but also in the links that can be maintained across successive (therefore non longer independent) realizations of the graph. Link persistence has a deleterious effect on community detection because it introduces *lagged inference* [35, 50], *i.e.*, the reconstruction at time $t$ accounts for the realization of the network at earlier than present time. Specifically, the following generative model is now assumed:

$$A_{ij}^{(t+1)} = \begin{cases} A_{ij}^{(t)} & \text{w.p. } \tau \\ \Delta_{ij}^{(t)} & \text{w.p. } (1-\tau) \end{cases} \quad \text{where} \quad \Delta_{ij}^{(t)} = \begin{cases} 1 & \text{w.p. } \theta_i\theta_j \frac{C_{\ell_{i_t},\ell_{j_t}}}{n} \\ 0 & \text{otherwise.} \end{cases} \tag{49}$$

The scenario covered in Section 2 of the main article allows one to infer the community structure from $\{\Delta^{(t)}\}_{t=1,...,T}$ but we only observe its "spoiled" version $\{A^{(t)}\}_{t=1,...,T}$. In order to overcome this limitation, we introduce the following matrix:

$$\tilde{A}_{ij}^{(t+1)} = \begin{cases} 1 & \text{if} \quad A_{ij}^{(t+1)} = 1 \quad \text{and} \quad A_{ij}^{(t)} = 0 \\ 0 & \text{else.} \end{cases} \tag{50}$$

In other words, if the same link is repeated at two successive time steps, it is deleted, because, if it was repeated, with high probability it must have been copied (recall that the probability of a link to spontaneously appear in our sparse regime is of order $O(1/n)$). Given the sparsity of $\Delta$, the matrices $\tilde{A}_{ij}^{(t+1)}$ and $\tilde{A}_{ij}^{(t)}$ are asymptotically independent and we thus recover the framework considered in Section 2 of the main article, when using $\tilde{A}^{(t)}$ (instead of $A^{(t)}$), provided that the detectability conditions on $\tilde{A}^{(t)}$ are met.

Let us investigate this detectability aspect. Starting from

$$\mathbb{P}(\tilde{A}_{ij}^{(t+1)} = 1) = \mathbb{P}(A_{ij}^{(t+1)} = 1|A_{ij}^{(t)} = 0)\big(1 - \mathbb{P}(A_{ij}^{(t)} = 1)\big) \tag{51}$$

we compute the value of $\mathbb{P}(A_{ij}^{(t+1)} = 1)$ recursively:

$$\mathbb{P}(A_{ij}^{(t+1)} = 1) = \tau\mathbb{P}(A_{ij}^{(t)} = 1) + (1 - \tau)\,\mathbb{P}(\Delta_{ij}^{(t+1)} = 1) \tag{52}$$

and thus, from time $t = 1$,

$$\mathbb{P}(A_{ij}^{(t+1)} = 1) = \sum_{m=1}^{t} \mathbb{P}\left(\Delta_{ij}^{(t+1-m)} = 1\right)\tau^m - \sum_{m=1}^{t-1} \mathbb{P}\left(\Delta_{ij}^{(t+1-m)} = 1\right)\tau^{m+1} = O_n\left(\frac{1}{n}\right). \tag{53}$$

Hence, injecting Equation 53 into Equation 51, we obtain

$$\mathbb{P}(\tilde{A}_{ij}^{(t+1)} = 1) = \mathbb{P}(A_{ij}^{(t+1)} = 1|A_{ij}^{(t)} = 0)(1 + o_n(1)) = (1 - \tau)\theta_i\theta_j\frac{C_{\ell_{i_t},\ell_{j_t}}}{n} + o_n(1). \tag{54}$$

The generative model of $\tilde{A}_{ij}^{(t)}$ thus asymptotically follows a DC-SBM in which the entries of $C$ are multiplied times $(1 - \tau)$.

To test our theoretical analysis, we evaluate numerically the percolation threshold and the detectability threshold on the matrix $\tilde{A}^{(T)}$. More specifically, the percolation threshold defines the condition under which the graph corresponding to $\tilde{A}^{(T)}$ has a giant component. For the DC-SBM (which generates $\Delta^{(T)}$), this condition is met whenever $c\Phi > 1$ [27]. The generative model of $\tilde{A}^{(T)}$ is asymptotically a DC-SBM in which all entries of the matrix $C$ are multiplied times a factor $(1 - \tau)$. The percolation threshold hence becomes

$$(1 - \tau)c\Phi > 1 \tag{55}$$

In the left plot of Figure 9 we generated, for different values of $c\Phi$ and different values of $\tau$, a sequence of $T = 15$ snapshots according to Equation (49) and plotted in color code the size of the giant component of $\mathcal{G}_T$, divided by the size of the graph. The dash-dotted black line indicates the position of the percolation threshold that evidences a good agreement between the theoretical prediction and the numerical experiment.

Concerning the detectability threshold, instead, the updated (static) detectability threshold here reads

$$\alpha > \frac{1}{\sqrt{1 - \tau}}. \tag{56}$$

In order to estimate $\alpha$ we compute $\hat{\alpha}$

$$\hat{\alpha} = \frac{\hat{c}_{\text{in}} - \hat{c}}{\sqrt{\hat{c}}}\sqrt{\hat{\Phi}}, \tag{57}$$

Figure 9: **Left**: Size of the biggest connected component divided by $n$ as a function of $c$ and $\tau$. The black line indicates the theoretical percolation threshold on the matrix $\tilde{A}$, Equation (55). **Right** Plot of $\mathrm{th}(3.5(\hat{\alpha}-1))$ (see text) as a function of $\alpha$ and $\tau$. The black line indicates the static detectability threshold on the matrix $\tilde{A}$, Equation (56). **For both figures**: $T = 15$, $n = 5\,000$, $k = 2$, $\Phi = 1.65$, $\eta = 0.8$. Averages are taken over three samples.

where

$$\hat{c} = \frac{1}{n} \sum_{i,j \in \mathcal{V}_T} \tilde{A}_{ij}^{(T)}; \quad \hat{c}_{\mathrm{in}} = \frac{2}{n} \sum_{i,j \in \mathcal{V}_T : \ell_i = \ell_j} \tilde{A}_{ij}^{(T)}; \quad \hat{\Phi} = \frac{1}{n\hat{c}^2} \sum_{i \in \mathcal{V}_T} \left( \sum_{j \in \mathcal{V}_T} \tilde{A}_{ij}^{(T)} \right)^2 \quad (58)$$

With a similar procedure as the one described to evaluate numerically the percolation threshold, in the right subplot of Figure 9, we display in color code the value of $\mathrm{th}(3.5(\hat{\alpha}-1))$, saturating the negative values to zero. When $\hat{\alpha} > 1$ the plotted function is between zero and one and we are above the transition. On the opposite, when $\hat{\alpha} < 1$ we are below the transition. The black dash-dotted line confirms the theoretical prediction of the detectability threshold, confirming also in this case our theoretical results.

Concluding, to get rid of the lag effect introduced by the persistence in the edges, one needs to remove at each time step the edges that are repeated. The positions of the information-theoretic transitions are asymptotically the same as those of a D-DCSBM model in which the entries of the matrix $C$ are re-scaled by a factor $1 - \tau$, the proportion of edges that do not get copied.

## E    Performance comparison

This section compares numerically the performance of Algorithm 1 against the main spectral methods commented along the paper. In Figure 10 the algorithms are tested for a different number of classes, value of $\eta$ and degree distribution. For $k > 2$ a symmetric setting with classes of equal size and $C_{ab} = c_{\mathrm{out}}$ for all $a \neq b$ is considered, so that the spectral algorithm of [14] is still well defined. Figure 10 indeed confirms that Algorithm 1 (i) benefits from high label persistence $\eta$; (ii) systematically outperforms the two considered competing dynamical sparse spectral algorithms [26], [14]; (iii) is capable of handling an arbitrary degree distribution.

To compare the performance of Algorithm 1 and the static Bethe-Hessian of [28], the case of small and large values of $\alpha$ should be treated separately. Close to the transition, Algorithm 1 improves over the static Bethe-Hessian and this gets more evident as $\eta$ increases: the joint solution of the problem at all times allows to improve the clustering performance in the hard detection regime. For large values of $\alpha$, instead, there seems to exist $\alpha^*(\eta)$ beyond which regularity only marginally improves the detection performance and Algorithm 1 performs equally (or slightly worse) than the static algorithm of [28]. Here, Algorithm 1 suffers the sub-optimal choices commented in Section 3 made to obtain a

Figure 10: Overlap comparison of Algorithm 1 (Alg 1), the dynamic adjacency matrix of [26] (Dyn A), the dynamic non-backtracking of [14] (Dyn B) and the static Bethe-Hessian of [28] (Static BH). The title of each row ant column indicates the values of $\eta, k, \Phi$ considered. For $\Phi \neq 1$ a power law degree distribution is adopted. The value of $\alpha$ is defined as $\alpha = \sqrt{c\Phi\lambda^2}$, where $\lambda = (c_{\text{in}} - c_{\text{out}})/(kc)$. The vertical line indicates the position of $\alpha/\alpha_c(T, \eta) = 1$. **For all simulations**: $c = 6$, $c_{\text{out}} = 0.5 \to 5$, $n = 25\,000$, $T = 4$. Averages are taken over 10 samples.

practical algorithm achieving non-trivial reconstruction when close to $\alpha_c(T, \eta)$. On the opposite, the static Bethe-Hessian of [28] is explicitly designed to optimally perform community detection for all values of $\alpha$ and any degree distribution, thereby justifying the two curves for large values of $\alpha$.

More specifically, Figure 11.A confirms that one can devise an optimal (but impractical) algorithm that exploits the eigenvector of $H_{\lambda, \eta}$ with null eigenvalue, as suggested in Section C.4. Close to the transition, the two dynamical methods perform similarly and largely outperform the static algorithm. For large values of $\alpha$, instead, Algorithm 1 suffers the sub-optimal (but practical) choice of $\xi = \lambda_d$, while for $\xi = \lambda$ the dynamical Bethe-Hessian is never beaten by the static Bethe-Hessian.
Figure 11.B instead compares the performance of Algorithm 1 with the dynamical adjacency matrix [26] and the static Bethe-Hessian [28] for a large value of $T = 25$, well evidencing the advantage of finding a joint solution of the clustering problem at all times.

A last remark concerns the capability of Algorithm 1 to recover communities of unequal sizes. Figure 11.C shows the accuracy of reconstruction of two communities of different size, as a function of the size of the smallest cluster over the size of the biggest. In order to obtain comparable results for different values of the ratio of the sizes of the two clusters, the following strategy is adopted: let $\Pi \in \mathbb{R}^{2 \times 2}$ be the diagonal matrix defined so that $\Pi_{ii}$ is the fraction of nodes belonging to class $i$ ($\text{Tr}(\Pi) = 1$). By imposing $C\Pi\mathbf{1}_2 = c\mathbf{1}_n$, the expected average $c$ is independent of the class label and it corresponds to the leading eigenvalue of $C\Pi$. The second eigenvalue of $C\Pi$, instead, determines the hardness of the detection problem (in the case of two classes of equal size it equal $(c_{\text{in}} - c_{\text{out}})/2$). For a given ratio $\Pi_{11}/\Pi_{22}$, the matrix $C$ is constructed so to let the leading eigenvalue of $C\Pi$ equal to $c$, and the second eigenvalue equal to a fixed value. For each time $t \geq 2$, the size of each class is

Figure 11: **A**: overlap comparison of Algorithm 1 (Alg 1), the static Bethe-Hessian of [28] (Static BH) and the reconstruction obtained using the eigenvector with zero eigenvalue of $H_{\lambda,\eta}$ (opt. BH dyn). For this simulation $n = 25\,000$, $k = 2$, $\Phi = 1$, $c_{\text{out}} = 0.5 \to 5$, $c = 6$, $T = 4$, $\eta = 0.9$. Averages are taken over 10 samples. **B**: overlap comparison for Algorithm 1 (Alg 1), the dynamical adjacency matrix of [26] (dyn A) and the static Bethe-Hessian of [28] (static BH) for large $T$. For this simulation $n = 500$, $k = 2$, $\Phi = 1$, $c_{\text{out}} = 2 \to 5.5$, $c = 6$, $T = 25$, $\eta = 0.8$. **C**: Overlap averaged over time achieved by Algorithm 1 on graphs with two communities of different size, as a function of the ratio of the size of the two communities. For this simulation $n = 10\,000$, $T = 5$, $c = 6$, $\Phi = 1$, $\eta = 0.7$. The second largest eigenvalue of $C\Pi$ is fixed to $s_2(C\Pi) = 4$. Averages over 15 samples.

kept fixed, by reassigning the labels according to the rule

$$\ell_{i_t} = \begin{cases} \ell_{i_{t-1}} & \text{w.p. } \eta \\ a & \text{w.p. } (1-\eta)\Pi_{aa}, \ \ a \in \{1,2\}. \end{cases} \tag{59}$$

The overlap (averaged over time) is then evaluated independently over the large and small class, to keep this measure meaningful: in the case $|\mathcal{V}_{\text{small}}| \gg |\mathcal{V}_{\text{large}}|$, assigning all nodes to the same cluster would output a large overlap.

## F  A fast implementation

A naive implementation of Algorithm 1 runs in $\mathcal{O}(nT \sum_{l=k}^{m} l^2)$ where $m$ is the *a priori* unknown number of negative eigenvalues of $H_{\lambda_d,\eta}$. Indeed, one (i) starts by computing the $k$ eigenvectors associated to the lowest eigenvalues of $H_{\lambda_d,\eta}$, costing $\mathcal{O}(nTk^2)$ via for instance classical restarted spectral Arnoldi algorithms [37]; (ii) verifies that the largest found eigenvalue is still negative; (iii) computes the $k+1$ eigenvectors associated to the lowest eigenvalues of $H_{\xi,\lambda_d}$; (iv) checks that the largest found eigenvalue is still negative; (v) iterates this process until the largest found eigenvalue crosses zero.

A much faster approximate implementation is described in Algorithm 2. The computation of the embedding $Y$ (line 10) should be done iteratively and thus costs $\mathcal{O}(pnT \log(nT))$, where $p$ indicates the order of the polynomial approximation $\tilde{f}$ (defined in line 8). The $T$ $k$-means steps cost $\mathcal{O}(nTk \log(nT))$. The overall cost is thus $\mathcal{O}(nTk \log(nT))$, where the constant $p$ is omitted as it is a problem-independent numerical factor.

To be complete, we recall here the two main arguments behind this accelerated algorithm: random projections and polynomial approximation. Further details may be found in [38, 39, 40].

**A preliminary observation.** Let $X \in \mathbb{R}^{nT \times m}$ be the exact eigenvectors of $H_{\lambda_d,\eta}$ associated to negative eigenvalues. They are obviously also the eigenvalues between 0 and $-\mu_{\min} > 0$ of the shifted matrix (used in Algorithm 2) $H'_{\lambda_d,\eta} = H_{\lambda_d,\eta} - \mu_{\min}I_{nT}$, where $\mu_{\min}$ is the smallest eigenvalue of $H_{\lambda_d,\eta}$. Algorithm 1 then performs $k$-means on the rows of $\{X_{i_t}\}_{i=1,\dots,n}$ for any $t = 1,\dots,T$. An important observation is that $k$-means only relies on the Euclidean distance between the feature

---

**Algorithm 2** A fast approximate implementation of Algorithm 1.

---

1: **Input** : adjacency matrices $\{A^{(t)}\}_{t=1,\dots,T}$ of the undirected dynamical graph $\mathcal{G} = \{\mathcal{G}_t\}_{t=1,\dots,T}$, label persistence $\eta$, number of clusters $k$; and parameters typically set to $p = 50$ (the order of the polynomial approximation) and $r = 10\log(nT)$ (the dimension of the random projection)
2: **for** $t = 1 : T - 1$ **do**
3:      Remove from $A^{(t+1)}$ the edges appearing in both $A^{(t)}$ and $A^{(t+1)}$ (Appendix D)
4: Compute $\lambda_d$ as in Algorithm 1 and create the dynamical Bethe-Hessian matrix $H_{\lambda_d,\eta} \in \mathbb{R}^{nT \times nT}$
5: Compute $\mu_{\min}$ and $\mu_{\max}$ the minimal and maximal eigenvalues of $H_{\lambda_d,\eta}$
6: Build $H'_{\lambda_d,\eta} = H_{\lambda_d,\eta} - \mu_{\min}I$, the shifted positive semi-definite version of $H_{\lambda_d,\eta}$.
7: Consider the step function $f(\mu) = 1$ if $\mu \leq -\mu_{\min}$ and $0$ if $\mu > -\mu_{\min}$.
8: Compute the coefficients $\{\alpha_k\}_{k=0,\dots,p}$ of the order $p$ Jackson-Chebychev polynomial approximation of $f$ on the interval $[0, \mu_{\max} - \mu_{\min}]$:

$$\forall \mu \in [0, \mu_{\max} - \mu_{\min}], \qquad f(\mu) \simeq \tilde{f}(\mu) = \sum_{k=0}^{p} \alpha_k \mu^k.$$

9: Generate a random matrix $R \in \mathbb{R}^{nT \times r}$ with iid Gaussian entries such that $\mathbb{E}(RR^T) = I$.
10: Compute $Y \in \mathbb{R}^{nT \times r}$ as

$$Y = \tilde{f}(H'_{\lambda_d,\eta})R = \sum_{k=0}^{p} \alpha_k H'^k_{\lambda_d,\eta} \, R$$

11: Normalize the rows of $Y_{i,:} \leftarrow Y_{i,:}/\|Y_{i,:}\|$
12: **for** $t = 1 : T$ **do**
13:      Estimate the community labels $\{\hat{\ell}_{i_t}\}_{i=1,\dots n}$ using $k$-class *k-means* on the rows $\{Y_{i_t}\}_{i=1,\dots,n}$.
14: **return** Estimated label vector $\hat{\boldsymbol{\ell}} \in \{1,\dots,k\}^{nT}$.

---

vectors $\boldsymbol{f_i} = X^T \boldsymbol{\delta_{i_t}} \in \mathbb{R}^m$, where the only non-zero entry of $\boldsymbol{\delta_{i_t}} \in \{0,1\}^{nT}$ is precisely $i_t$,

$$d_{ij}^2 = \|\boldsymbol{f_i} - \boldsymbol{f_j}\|_2^2. \tag{60}$$

As such, $k$-means does not *need* the exact matrix $X$, but rather only feature vectors whose interdistances verify the above. The random projections discussed in the next paragraph aim at creating random feature vectors whose interdistances concentrate around the above Euclidean distance.

**Random projection.** Denote by $R \in \mathbb{R}^{nT \times r}$ a random matrix with for example Gaussian i.i.d. entries verifying $\mathbb{E}(RR^T) = I$. Define $Y = XX^T R \in \mathbb{R}^{nT \times r}$ and new feature vectors $\bar{\boldsymbol{f}}_i = Y^T \boldsymbol{\delta_i} \in \mathbb{R}^r$. One has, denoting $\boldsymbol{\delta_{ij}} = \boldsymbol{\delta_i} - \boldsymbol{\delta_j}$:

$$\forall i,j \qquad \bar{d}_{ij}^2 = \|\bar{\boldsymbol{f}}_i - \bar{\boldsymbol{f}}_j\|_2^2 = \|R^T XX^T \boldsymbol{\delta_{ij}}\|_2 \tag{61}$$

and in expectation:

$$\begin{aligned}
\forall i,j \qquad \mathbb{E}\left(\bar{d}_{ij}^2\right) &= \mathbb{E}\left(\boldsymbol{\delta_{ij}^T} XX^T RR^T XX^T \boldsymbol{\delta_{ij}}\right) \\
&= \boldsymbol{\delta_{ij}^T} XX^T \mathbb{E}\left(RR^T\right) XX^T \boldsymbol{\delta_{ij}} \\
&= \boldsymbol{\delta_{ij}^T} XX^T XX^T \boldsymbol{\delta_{ij}} \\
&= \boldsymbol{\delta_{ij}^T} XX^T \boldsymbol{\delta_{ij}} \\
&= d_{ij}^2.
\end{aligned}$$

Importantly, the concentration of the expectation around its expected value is *fast*. The Jonhson Lindenstrauss lemma states that $r = \mathcal{O}(\frac{1}{\epsilon^2}\log nT)$ suffices for a $(1+\epsilon)$ multiplicative approximation of the Euclidean distance (see [38, 39] for a lengthier discussion).

**Polynomial approximation.** In our context, these random projections are pointless as long as we do not have an efficient way to obtain $Y$ without actually computing $X$. This problem can be solved using a polynomial approximation. Let us write the diagonalized form of $H'_{\lambda_d,\eta}$ as $H'_{\lambda_d,\eta} = U\Lambda'U^T$ where $\Lambda'$ is the diagonal matrix of eigenvalues $\{\mu'_i\}$. Let us write the matrix

Figure 12: **Top**: computation time of Algorithm 2 versus the number of nodes $n$ (left, for $T = 2$), the number of timesteps $T$ (middle, for $n = 1000$) and the number of communities $k$ (right, for $n = 1000$), with parameters $\eta = 0.7$, average degree $c = 6$, $\Phi = 1.6$, and $\alpha = 1.5\alpha_c(T, \eta)$. **Bottom**: performance comparison between Algorithm 1 and Algorithm 2 in terms of (left) overlap versus $\alpha/\alpha_c(T, \eta)$ for $n = 5000$, $T = 10$, $k = 2$, $c = 6$, $\eta = 0.5$, $\Phi = 1.6$ and in terms of (right) computation time versus $T$ for $n = 300$, $k = 2$, $c = 6$, $\alpha = 1.5\alpha_c(T, \eta)$, $\eta = 0.5$, $\Phi = 1.6$. On all figures, the results are the average over 40 experiments.

function $f(H'_{\lambda_d,\eta}) = U f(\Lambda') U^T$ for any function $f$ defined on the spectrum of $H'_{\lambda_d,\eta}$. Let us consider the particular step-function function $f(\mu)$ that is equal to 1 if $\mu \leq -\mu_{\min}$ and to 0 if $\mu > -\mu_{\min}$. Note that $XX^T = f(H'_{\lambda_d,\eta})$.

Define $\tilde{f}(\mu) = \sum_{k=0}^{p} \alpha_k \mu^k$ a polynomial approximation of order $p$ of $f(\mu)$ on the interval $[0, \mu_{\max} - \mu_{\min}]$ (the larger $p$ the better the approximation). One can compute an approximation of $Y$ using $\tilde{f}$:

$$Y = f(H'_{\lambda_d,\eta}) \, R$$

$$\simeq \tilde{f}(H'_{\lambda_d,\eta})R = U \sum_{k=0}^{p} \alpha_k \Lambda'^k U^T R = \sum_{k=0}^{p} \alpha_k H'^k_{\lambda_d,\eta} \, R.$$

The choice of which polynomial approximation to choose is not straightforward. One possible choice is to use Chebychev polynomials as they have a guarantee on the infinite norm of the approximation error. However, they tend to create Gibbs oscillation around sharp cut-offs of the function to approximate. As the function we wish to approximate here is a step function, it is customary to choose Jackson-Chebychev polynomials (which explicitly dampen these unwanted oscillations). See discussions in [40, 51, 52].

**In practice.** Fig. 12 (top) experimentally illustrates that the complexity of Algorithm 2 is indeed linear in $n$, $T$ and $k$. The bottom of Fig. 12 compares both Algorithms in terms of overlap and computation time: Algorithm 2, being only an approximation, never performs as well as Algorithm 1, especially as the detection problem becomes more difficult and the control parameter $\alpha$ approaches the transition point $\alpha_c$. However, the gain in computation time is drastic as $m$ increases (here $k$ is fixed to 2 and $T$ increases).

## Footnotes

[8]Precisely, quoting the authors, this is as far as $\alpha_c(T, \eta)$ is defined: "We can compute the corresponding finite-time threshold for a fixed $T$ by diagonalizing a $(3T - 2)$-dimensional matrix, where we have a branching process with states corresponding to moving along spatial, forward-temporal, or backward-temporal edges at each time step".

[9]In this fixed point the messages are independent of the class labels, hence it is called *trivial*. From a simple substitution one can indeed verify that it is a fixed point.