[Reviews · NeurIPS 2020]

Review 1

Summary and Contributions: In this paper, the authors extend via a straight forward derivation the conjecture of detectability threshold of community detection problem, for k=2 (number of clusters), and for a network generated using a dynamic degree-corrected stochastic blockmodel with link persistency. One contribution of the paper is an extension of the results previously conjectured in Ref. [14] for dynamic SBM to dynamic DC-SBM using the results from [12,13]. Also the main contribution of the paper is the extension of Bethe Hessian spectral clustering as a dynamic spectral clustering on temporal networks. -------post author response ---------- Thanks to the authors for their clarifications. After reading the authors' feedback, I have still serious concerns regarding the misunderstanding that can be created by this paper regarding Ref. [14]. If the authors have access to the codes by Ref. [14] that seems they have because of the footnote [8], then why they didn't provide the corresponding simulation with dynamical non-backtracking of [14]. To be fair to the readers and also to authors in Ref. [14], it would be correct that authors make sure to update their figure 3 with the regularized algorithm in Ref. [14] that seems have similar results with Alg. 1, instead of mentioning provided results is just for the second eigenvector, when it is obvious this eigenvector is not informative. Regarding the novelty of the paper I still think this paper is not in the level of a NeurIPS paper since the contributions are partial and more importantly if the authors have been provided the results of Ref. [14] in Fig 3 in the first place, then the reviewers could judge the performance of their proposed spectral algorithm compared to existing methods. Although as I said before, the paper is really well-written and then I decided to increase my score. ------------------------------------------

Strengths: The paper is well written and technically sounds.

Weaknesses: Although the paper is well written and explained well, however, I think the novelty of the paper is not enough. More importantly, a serious concern regarding the issues with some of the results previously shown in Ref. [14] that is not consistent with the results in this paper. I have enumerated some of the issues I found in the publication that I have explained below. Some comments: 1. In page 3, line 98, the authors explain a method in the main text regarding how to derive the detectability limits for finite T, however, this is explained in Ref. [14]. I think the authors need to explain it in the "main text" that this is an explicit rederivation of what the authors in [14] explained before and not just in the appendix. 2. Line 152-153: The authors say "Letting \omega_{ij}=\xi if there exists a time instant t such that nodes i, j belong to Vt , and \omega_{ij} = h otherwise, ...". I think the authors meant \omega_{ij} = h for temporal edges, but it is needed to be written more clearly unless it means for all non-edges. Please make a correction. 3. The main concern is regarding the simulation of dynamical non-backtracking. The main contribution of the paper is that they show that the spectral method they propose is the one that detect communties as soon as theoretically possible and no spectral algorithm can do that, however, previously this has been shown in Ref. [14], where the authors show detectability is possible all the way down to the detectability threshold. Ref. [14] says "We then give two algorithms that are optimal in the sense that they succeed all the way down to this threshold. The first uses belief propagation, which gives asymptotically optimal accuracy, and the second is a fast spectral clustering algorithm, based on linearizing the belief propagation equations". It seems the authors's simulation for the proposed algorithm in Ref. [14] is problematic for simulation of dyn B method and it seems they have chosen an uninformative eigenvector for reconstruction (overlap is 0 !). See Fig. 3 in Ref. [14]. 4. The results are limited for K=2 and I am not sure why the authors that provided the algorithm specifically for dynamic DC-SBM didn't provide more results for K>2, different values of \Phi, and larger T (T=4 is really small for a dynamic network). some typos: 1. line 147: "whose theoretical support in given" --> "... is given" 2. line 519: "2.2" --> "3.2"

Correctness: The algorithms seems valid for K=2, however, the results are not provided for K>2. Also one of the algorithms that the authors compared with i.e. the dynamical non-backtracking Ref. [14] is wrongly simulated. See Fig. 3 in Ref. [14].

Clarity: Yes, well-written

Relation to Prior Work: Yes, it is clearly explained

Reproducibility: Yes

Additional Feedback:


Review 2

Summary and Contributions: The main contribution of this paper is the the development and analysis of a dynamic version of the Bethe-Hessian matrix-based algorithm for community detection, which also performs well in the sparse regime.

Strengths: The topic of this paper pertains to temporal networks, an area of growing interest in the recent years. The paper is clearly structured, and the supplementary material provides comprehensive additional details on the main steps of the proposed pipeline. The proposed model is a sensible one, where a fraction of the nodes maintain their respective cluster membership, while the remaining ones receive random cluster re-assignments, and at each time step the adjacency matrix follows a degree corrected stochastic block model. The authors perform the sanity check that higher label persistence lead to being able to solve harder problems (sparser and more noise). The experiments in Figure 3 show superior performance to other methods from the literature, slightly under than if Belief Propagation algorithm, which however, is 2-3 orders of magnitude more computationally expensive for the problem instance considered.

Weaknesses: The authors could make it more clear from the outset under what sparsity regime specifically they are able to operate in (in terms of average degree or density). Figure 3 Left is not clear and is missing titles for the subplots and the quantity being heatmapped. The Figure caption is also missing k (=2 I presume). Authors should clarify at the beginning of Section what is meant by overlap performance, perhaps less of a conventional name. Why not stick to something ubiquitous like Adjusted Rank Index? For the numerical simulation in the appendix, concerning k, it might be interesting to see what empirically happens for unequal cluster sizes (preferably unequal in expectation), especially as a function of the ratio between the smallest and largest cluster (in other related settings of SBM models, the position of the informative eigenvalues may change). Can the authors comment on the extension of this approach to directed graphs (where the adjacency matrix may or may not be assumed skew-symmetric)? Minor typos: Line 51/255: run — > ran?

Correctness: The paper appears to be technically sound.

Clarity: Yes, the paper is clearly written and well structured.

Relation to Prior Work: Yes, the authors explain how this work relates to existing literature on clustering temporal graphs, and what gaps it aims to fill.

Reproducibility: Yes

Additional Feedback: ----- Post Rebuttal ----- After reading the rebuttal and the other reviewers’ comments, I still believe this is solid submission for NeurIPS. I have lowered my score in light of the issues pointed out in relation to Ref [14].


Review 3

Summary and Contributions: The paper studies community detection in sparse dynamical graphs with the dynamical degree-corrected stochastic block model. It aims to address both sparsity and the so-called small label persistence issues by proposing a new spectral algorithm. Specifically, a dynamical Bethe-Hessian matrix is introduced. Authors claim that it can retrieves non-trivial communities for the case that only two communities exist (they also give an algorithm that can handle the case of more than two communities). Theoretical analysis and experimental studies are provided. UPDATE: I have read the rebuttal.

Strengths: The paper seems to make some theoretical contribution to tackling the sparsity and small label persistence issues in dynamical graphs.

Weaknesses: There is no case study on real dynamical graphs. The proposed algorithms seem far away from being deployed in practice. I feel the paper is difficult to read, probably because I am not familiar with the context of statistical physics favor.

Correctness: seem correct

Clarity: yes, but it could be improved for broader readers.

Relation to Prior Work: yes

Reproducibility: Yes

Additional Feedback:


Review 4

Summary and Contributions: The paper introduces a spectral clustering algorithm for a generalization of the degree-corrected stochastic blockmodel to a dynamic setting in which the groups labels evolve over time (are kept) according to a particular "persistence" dynamics in which the group label is either kept or randomly reassigned. The authors generalize the ideas of spectral clustering via the Bethe Hessian to this dynamical setting and provide some theoretical discussion and analysis of their algorithms, in particular by relating their techniques to the (dynamic variant) of the backtracking matrix. Overall, I find this study to be quite interesting and a welcome addition to the current literature on spectral clustering.

Strengths: - The presented method has a clear grounding in spectral clustering schemes that have been developed for static graphs, and the authors do a good job in revisiting these foundations and appropriate extensions. - spectral clustering is a very widely used technique, and I think there is a clear significance of extending these results to the dynamic setting

Weaknesses: - As this work heavily draws on previous research on the Bethe Hessian for static graphs, the novelty is somewhat limited; in particular as the author can not give (due to the inherent additional complexity of the dynamical setting) not always give definitive algorithms and analysis, and have to make additional assumptions (e.g., knowing the persistence rate \eta) -The empirical validation is somehwat limited, e.g., some application to real data would have strongly increase the impact of this work and there is a large number of temporal netwok data available by now (the auhtors provide some citations).

Correctness: The claims made by the authors are by and large supported by theoretical analysis, even though the arguments used are not always fully rigorous but appeal to physcially motivated conjectures and/or previous literature on the subject.

Clarity: The paper is written in a clear and cohesive manner

Relation to Prior Work: The relations to previous work are appropriately highlighted in my opinion. However, the novelty of their approach w.r.t to previous works could be discussed more clearly.

Reproducibility: Yes

Additional Feedback: p4. line 125 : \sigma should be s ? p5. line 132: needless -> useless / non informative? p.7, line 237: a closing parenthesis is missing p.8, line 277: "we hinted at a greedy line-search -> maybe I misunderstood some of the remarks in section 4, but it appears there is (if anything) only a very subtle hint at a line search procedure in section 4?! Following discussions and author response, I still have some reservations here, but see the paper slightly above the threshold.

[Author Response · NeurIPS 2020]

We thank the reviewers for the insightful comments.

**On the novelty of our approach (R1,R4)**. As it is for [14], our article takes inspiration from the works considering
the problem in the static regime [9,18,19,28] that, although not formally rigorous, all converge to similar conclusions,
recently partially proved in *e.g.* [10-13]. To the best of our knowledge, the proposed article and [14] provide the
only methods capable of making non-trivial reconstruction as soon as theoretically possible. However, as opposed to
[14], in our article we study much more in depth the spectrum of the matrix $B$, consequently proposing a dynamical
Bethe-Hessian that, compared to $B$, is smaller in size and symmetric, hence allows for a faster computation of the
relevant eigenvectors. The choice $\xi = \lambda_d$, introduces further advantages: i) the value of $\lambda$ (needed to build $B$) does not
need to be known; ii) as claimed in the article, Algorithm 1 is well defined for $k \geq 2$ classes of *arbitrary size* (unlike
[14]). These aspects suggest that Algorithm 1 should be preferred over the spectral algorithm of [14]. In agreement with
the points raised by reviewers 2 and 4, given the growing interest in dynamical networks and the vast use of spectral
clustering, our submission has the potential to strongly impact the field of dynamical community detection.

**Simulation of [14] (R1)**. After a personal communication with the authors of [14], it appears that the algorithm
deployed in [14] to obtain Figure 3 uses a regularization of $B$ to suppress the *uninformative* modes and locate the
informative eigenvalues in dominant position. This regularization step unfortunately is not mentioned in the published
version of [14] and the authors are currently at work to clarify this point. We eventually learned of the existence of an
unpublished version of [14] (arXiv:1506.06179) in which the authors mention the need of a regularization and show in
Figure 2 that the spectrum of $B$ for $k = 2$ can indeed have more than two isolated eigenvalues. In Figure 3 we show the
performance of $B$ when using the second largest eigenvector, as prescribed in [14]. We further underline that the results
of our Appendix C imply that the matrix $B$ can make non-trivial partitions for $\alpha > \alpha_c(T, \eta)$, in agreement with the
conjectures of [14]. To make this point clearer, we will move the simulation of *opt. dyn B* of Figure 10 to Figure 3.

**Comprehensiveness of the simulation on synthetic data (R1, R2)**. Our theoretical case study was limited to two
classes of equal size, typical setting used in the literature (see e.g. [18,19]) for which the transition is well defined. For
consistency, we chose to show only the most significant case of $k = 2$ in the main article and explore the behavior of
the algorithm for different values of $\eta, k, \Phi$ in the Appendix. The reviewers suggested to further simulate the behavior
of the algorithm for larger values of $T$ and for classes of different sizes. Algorithm 1 is well defined also in these cases
and relevant simulations will be added to the Appendix showing that: i) the performance increases as $T$ increases; ii)
Algorithm 1 can reconstruct communities of very different sizes (ratio 9/1).

**Applicability and analysis on real data-sets. (R3, R4)** For any given temporal graph and a value of $\eta$, Algorithm 1 is
well defined. Other notable works in the static regime [18,19,28] showed that the non-backtracking and Bethe-Hessian
matrices perform well also for graphs not generated from the DC-SBM and we expect the same to hold for Algorithm
1. If $\eta$ is not known, as we indicate in lines 219-225, it can be selected through cross validation as the value leading
to the largest number of classes detected. This aspect will be clarified in the updated version and it makes Algorithm
1 applicable to any temporal graph. As suggested, we tested its performance on some real datasets taken from the
SocioPattern webpage (*e.g.* Primary, LyonSchool) observing that: i) for the same $k$ it outperforms [26] in terms of
modularity; ii) for similar modularity values, it outputs an estimate of $k$ much closer to the ground truth than [41].

**Explicit derivation of $\alpha_c(T, \eta)$ (R1).** Appendix A consists of an explicit re-derivation of the results of [14], generalized
to the case of $\Phi \neq 1$. This is clearly stated in the Appendix, but it certainly deserves to be properly addressed in the
main text as well. A clear comment will be added in the new version.

**Definition of $\omega_{ij}$ for the temporal edges (R1).** As per Definition 1 and lines 151-153, the non-edges do not take part
in the definition of $B_{\xi,h}$ and the weight $\omega_{ij} = h$ effectively only concerns the time edges. We will clarify this definition.

**Regime of sparsity considered (R2).** The regime of sparsity in which we are able to operate is an implicit consequence
of the detectability condition. In fact, Algorithm 1 can be adopted for all $c_{in}, c_{out} = O_n(1)$ that satisfy $\alpha > \alpha_c(T, \eta)$
and that guarantee the existence of giant component (as stated in the footnote 3). We will make this aspect more explicit.

**Figure 3 (R2).** The overlap is defined in footnote 7. For clarity the definition might be moved to the main text. The
overlap expresses how much a label assignment performs better than random guess, relatively to the planted assignment.
It ranges between 0 (random assignment) to 1 (perfect assignment). We chose this measure that is adopted in many
other works in the field, *e.g.* [14,18,19,28]. The left plot is a representation of the overlap (heatmapped) against $\alpha$ and
$\eta$, while the right plot is a horizontal cut of the left plot at $\eta = 0.7$ for different techniques. For both plots, $k = 2$.

**Extension to non-symmetric matrices (R2).** The approach based on the Bethe-Hessian matrix can be easily general-
ized to weighted and directed graphs, as initially suggested in [19]. The use of a different generative model (giving
a non symmetric matrix), however, would affect most of the analytical results obtained and the crucial aspect of the
detectability threshold. The extension of the concept of detectability threshold and of Proposition 1 would need a
dedicated analysis, but we do not predict any major complication in this extension.

[Meta-Review · NeurIPS 2020]

This paper provides some extensions on detectability of community structure in dynamic networks. There was substantial discussion of the contributions with respect to Reference [14] (Ghasemanian et al., PRX 2016 / arXiv:1506.06179). In the end, there was consensus that the contributions from this paper (both theoretical and algorithmic) are still meaningful, and so the recommendation is to accept. However, the referees and AC strongly encourage the paper to make the following changes for a camera version: 1. Give proper credit to the ideas in the arXiv version of Reference [14] and provide a fair comparison to the existing ideas in Figure 3 (see Reviewer 1’s suggestions). 2. Include the results on real-world data that were summarized in the author response.